# The barley immune receptor *Mla* recognizes multiple pathogens and contributes to host range dynamics

Jan Bettgenhaeuser [1,6,11], Inmaculada Hernández-Pinzón [1,11], Andrew M. Dawson[1], Matthew Gardiner[1], Phon Green [1], Jodie Taylor[1], Matthew Smoker[1], John N. Ferguson[1,7], Peter Emmrich[1,8], Amelia Hubbard[2], Rosemary Bayles[2], Robbie Waugh [3], Brian J. Steffenson [4], Brande B. H. Wulff [1,9], Antonín Dreiseitl [5], Eric R. Ward[1,10] & Matthew J. Moscou [1✉]

Crop losses caused by plant pathogens are a primary threat to stable food production. Stripe rust (*Puccinia striiformis*) is a fungal pathogen of cereal crops that causes significant, persistent yield loss. Stripe rust exhibits host species specificity, with lineages that have adapted to infect wheat and barley. While wheat stripe rust and barley stripe rust are commonly restricted to their corresponding hosts, the genes underlying this host specificity remain unknown. Here, we show that three resistance genes, *Rps6*, *Rps7*, and *Rps8*, contribute to immunity in barley to wheat stripe rust. *Rps7* cosegregates with barley powdery mildew resistance at the *Mla* locus. Using transgenic complementation of different *Mla* alleles, we confirm allele-specific recognition of wheat stripe rust by *Mla*. Our results show that major resistance genes contribute to the host species specificity of wheat stripe rust on barley and that a shared genetic architecture underlies resistance to the adapted pathogen barley powdery mildew and non-adapted pathogen wheat stripe rust.

[1] The Sainsbury Laboratory, University of East Anglia, Norwich Research Park, Norwich NR4 7UK England, UK. [2] NIAB, 93 Lawrence Weaver Road, Cambridge CB3 0LE England, UK. [3] The James Hutton Institute, Invergowrie, Dundee DD2 5DA Scotland, UK. [4] Department of Plant Pathology, University of Minnesota, St. Paul, MN 55108, USA. [5] Department of Integrated Plant Protection, Agrotest Fyto Ltd, Havlíčkova 2787, CZ-767 01 Kroměříž, Czech Republic. [6]Present address: KWS SAAT SE & Co. KGaA, 37574 Einbeck, Germany. [7]Present address: Department of Plant Sciences, University of Cambridge, Downing Street, Cambridge CB2 3EA, UK. [8]Present address: John Innes Centre, Norwich Research Park, Norwich NR4 7UH, UK. [9]Present address: Center for Desert Agriculture, Biological and Environmental Science and Engineering Division (BESE), King Abdullah University of Science and Technology, Thuwal 23955-6900, Saudi Arabia. [10]Present address: AgBiome, Research Triangle Park, NC 27709, USA. [11]These authors contributed equally: Jan Bettgenhaeuser, Inmaculada Hernández-Pinzón. ✉email: matthew.moscou@tsl.ac.uk

Plants are constantly exposed to a diverse array of pathogens, but remain resistant to the vast majority of microbial invaders[1]. This phenomenon is termed nonhost resistance[2] and collectively refers to the interaction of plants with non-adapted pathogens. Early work by Eriksson[3] in 1894 found that *Puccinia striiformis* (stripe rust) only infected Triticeae species from which they were originally isolated. This host species specificity appeared to be common among diverse rust pathogens of grasses. These specialized forms, designated *formae speciales*, were later observed in several filamentous pathogen systems, such as the powdery mildews of cereals (*Blumeria graminis*)[4] and smuts (*Ustilago* spp.)[5] on the Triticeae, vascular wilt caused by *Fusarium oxysporum*[6], blast (*Magnaporthe* spp.) of grasses[7], the downy mildews (*Bremia* spp.) of Asteraceae[8], and white rusts (*Albugo* spp.) of Brassiceae[9]. From these early observations of host adapted lineages in diverse plant-pathogen systems, considerable progress has been made in understanding the genetic architecture underlying resistance to non-adapted pathogens. One of the earliest approaches to discover the genetic basis of non-adapted resistance of rye (*Secale cereale* L.) involved the cytogenetic transfer of individual chromosomes to wheat (*Triticum aestivum* L.)[10]. Using interspecific crosses, Riley and Macer found that individual chromosomes from rye conferred resistance to wheat powdery mildew and wheat stripe rust. This work established that resistance to non-adapted pathogens in closely related species is genetically simple, likely determined by individual loci. Subsequent work using crosses of different *formae speciales* of *B. graminis* found that four resistance genes and avirulence genes conditioned the *formae speciales* divide between wheat and *Agropyron* and their corresponding powdery mildews[11]. While several genetic loci have been defined, it is unclear whether the resistance conditioned at these loci is distinct or overlapping with resistance to host pathogens.

The plant immune system is composed of successive layers of passive and active barriers that provide resistance to pathogens[12]. Passive barriers include the cuticle and cell wall, whereas active barriers include reactive oxygen species, secondary metabolites with antimicrobial activity, lignification of the cell wall, callose deposition, and cell death. Recognition of pathogens is mediated by membrane-bound and intracellular immune receptors, which serve to recognize pathogen-derived molecules and perceive danger signals[13,14]. Membrane-bound immune receptors (pattern recognition receptors) are structurally composed of a variable extracellular, ligand-binding domain, transmembrane domain or glycophosphatidylinositol (GPI) anchor, and a variable intracellular protein kinase domain[15]. The largest class of intracellular immune receptors in land plants are the nucleotide binding-leucine rich repeat (NLR) proteins[16]. Pathogens have evolved several mechanisms to overcome plant immunity. These include the use of enzymatic degradation and mechanical turgor to breach the cell wall[17], apoplastically secreted proteins that sequester fungal cell wall fragments that would elicit immune responses[18], and proteinacious and non-proteinacious effectors that are secreted into the plant cell to suppress immunity and facilitate nutrient acquisition[19].

NLR encoding loci are highly complex, exhibiting substantial copy number variation, structural rearrangements, and novel gene content between allelic variants[20]. Recognition of plant pathogens by NLR occurs through direct or indirect recognition of effectors[21]. Direct recognition occurs through a physical interaction of NLR and effector, whereas indirect recognition occurs through the monitoring of a host protein by an NLR. Upon modification of the host protein, the NLR initiates an immune response that leads to resistance. The mechanism underlying indirect recognition, i.e. guarding of host proteins by NLRs, suggests that multiple pathogens could be recognized if their effectors target a conserved host protein. Of the hundreds of described NLRs conferring resistance to plant pathogens, only a few NLRs have been shown to natively recognize taxonomically distant pathogens: *RPS4*/*RRS1*[22–25] and *Mi*[26,27]. The observed low frequency of NLRs that recognize multiple pathogens can be explained by several factors. It may be that NLRs are extremely specific in their interaction with host proteins such that they have the capacity to only recognize a specific modification or alternatively, an insufficient number of plant-pathogen systems have been investigated to establish the broader capacity for NLR recognition.

The *Mla* (*Mildew locus a*) locus[28] in barley (*Hordeum vulgare*) exhibits extensive structural and copy number variation, and is known to contain three different NLR encoding gene families (*RGH1*, *RGH2*, and *RGH3*)[29–32]. Within the locus, resistance to barley powdery mildew (*B. graminis* f. sp. *hordei*; *Bgh*) is conferred by members of the *RGH1* gene family. *Mla* has over 30 characterized alleles with different specificities for recognition of barley powdery mildew[33,34]. In closely related cereals such as rye (*Secale cereale*), the *Mla* homolog *Sr50* confers resistance to wheat stem rust (*Puccinia graminis* f. sp. *tritici*)[35], whereas *TmMla1*[36] and *Sr33*[37] in *Triticum monococcum* and *Aegilops tauschii* (wheat D-genome progenitor) confer resistance to wheat powdery mildew (*B. graminis* f. sp. *tritici*) and wheat stem rust (*P. graminis* f. sp. *tritici*), respectively. Several effectors recognized by the *Mla* (*RGH1*) gene family have been identified[38–40]. All of these are predicted small (102-132 aa) secreted peptides with divergent sequence composition, with the exception of the allelic variants $AVR_{a10}$ and $AVR_{a22}$[40]. Thus, *Mla* homologs recognize effectors from ascomycete and basidiomycete pathogens.

Stripe rust (yellow rust; *P. striiformis*) is a filamentous fungal pathogen of the Triticeae, causing substantial global yield loss, particularly in wheat[41]. After landing on a leaf, *P. striiformis* generates a germ tube, invades via stomata, and initiates the infection process through hyphal colonization, followed by a transition to pustule formation (reproduction). Wheat stripe rust (*P. striiformis* f. sp. *tritici*, *Pst*) and barley stripe rust (*P. striiformis* f. sp. *hordei*, *Psh*) exhibit host species specificity to wheat and barley. We set out to understand the genetic architecture underlying resistance to wheat stripe rust in barley (*Hordeum vulgare*). Previous work suggests that host species specificity to stripe rust is conditioned by resistance genes[42–46], but the complexity and diversity of these genes in a range of germplasm, their identity, and the impact of domestication and plant breeding on their prevalence remain unclear. In this study, we interrogate a representative panel of barley accessions for their reaction to this non-adapted pathogen and found three loci, *Rps6*, *Rps7*, and *Rps8*, which determine host species specificity in barley at different stages of the pathogen lifecycle. These loci are present across the range of barley diversity including wild, landrace, and elite accessions and form the basis of a wheat stripe rust resistance gene complex in barley. *Rps7* is mapped to the barley powdery mildew resistance locus, *Mla*. Based on high-resolution recombination screens and transgenic complementation we show that different alleles of *Mla* confer resistance to barley powdery mildew and wheat stripe rust.

## Results

**Wild barley exhibits extensive phenotypic diversity to *Pst* isolate 08/21 compared to domesticated barley.** We started our investigation of the genetic architecture of resistance to wheat stripe rust in barley by screening the reaction to *Pst* isolate 08/21 in a diverse collection of barley germplasm, which included 76 cultivars, 20 landraces, and 27 wild accessions (Supplementary Data 1)[47]. The genetic diversity of the panel was assessed by

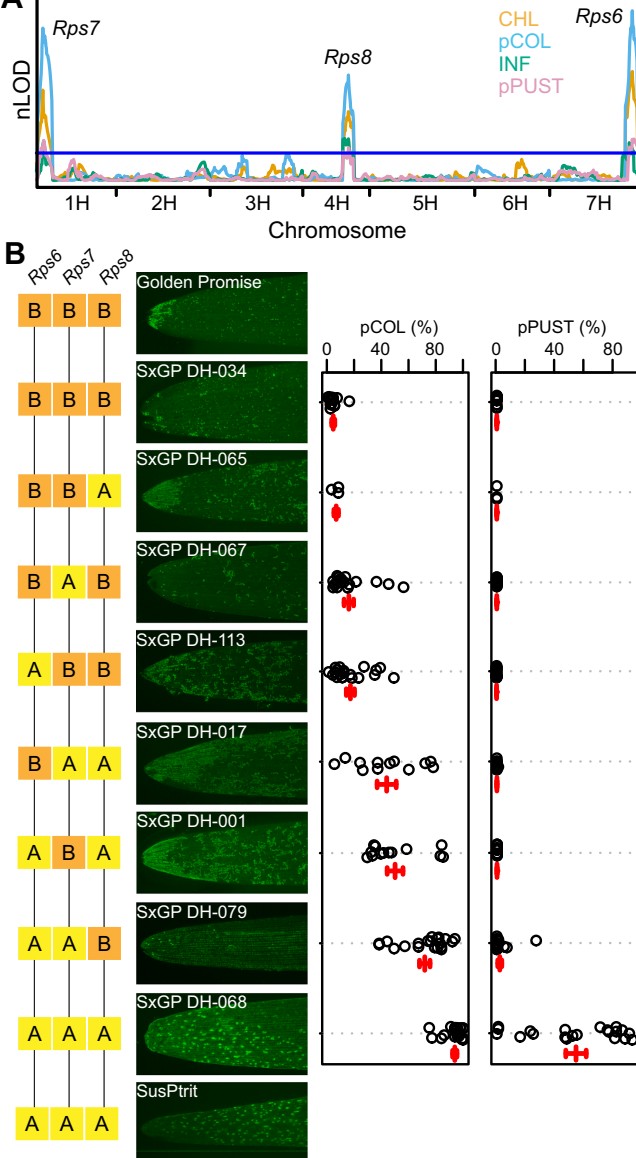

**Fig. 1 Three loci underpin resistance to wheat stripe rust (*P. striiformis* f. sp. *tritici*) isolate 08/21 in Golden Promise. A** Composite interval mapping of the SusPtrit × Golden Promise (SxGP) doubled-haploid (DH) population scored 14 days post-inoculation with *P. striiformis* f. sp. *tritici* isolate 08/21 using an additive model ($H_0$:$H_1$). Phenotypes include macroscopic chlorosis (CHL) and infection (i.e. pustule formation; INF) and microscopic hyphal colonization (pCOL) and pustule formation (pPUST). Results were plotted using normalized permutation thresholds (nLOD) with a threshold of statistical significance based on 1,000 permutations (blue horizontal line). **B** Three-way phenotype × genotype plots for pCOL and pPUST show the contribution of *Rps6*, *Rps7*, and *Rps8* to resistance to *P. striiformis* f. sp. *tritici* isolate 08/21 in the SxGP DH population. Error bars are mean and standard deviation for each genotypic group. 'A' and 'B' denote SusPtrit and Golden Promise alleles for each of the lines according to markers SCRI_RS_155652, BOPA2_12_30817, and BOPA1_4361–1867 for *Rps6*, *Rps7*, and *Rps8*, respectively. Micrographs are SxGP DH representative lines stained with WGA-FITC. Accessions and individuals per haplotype group: DH-034 ($N = 13$), DH-065 ($N = 3$), DH-067 ($N = 18$), DH-113 ($N = 20$), DH-017 ($N = 12$), DH-001 ($N = 13$), DH-079 ($N = 20$), and DH-068 ($N = 21$). Data shown are one of two replicates based on 122 DH lines, the same QTLs were identified in replicate experiments. Source data are provided as a Source Data file.

genotyping 129 accessions with 1,536 SNPs[48]. A total of 1,258 SNPs with a minor allele frequency greater than 5% and less than 10% missing data were selected and used to build a dendrogram using hierarchical clustering. From this analysis we observe four distinct groupings largely reflecting row and improvement status (Supplementary Fig. 1a). *P. striiformis* infects the leaves of cereals in a stepwise infection process that begins with intercellular colonization of leaves and then transitions to pustule formation[49]. Phenotyping was carried out using macroscopic chlorosis and infection on a nine-point scale (0 to 4, 0.5 increments) that correlated with the surface area exhibiting yellowing or pustule formation, respectively[47]. Cultivar and landrace barley accessions were highly resistant to wheat stripe rust. We observed that two-row elite barley accessions were highly resistant to *Pst* with only one accession exhibiting minor pustule formation. Only 14 accessions (22%) in this group displayed a chlorotic phenotype and no leaf was observed with greater than 50% chlorosis. The observation of near complete resistance to *Pst* among the two-row elite accessions contrasted with an increased occurrence of susceptibility in the wild accessions. Of the 27 accessions in the wild barley group, 23 (85%) exhibited varying degrees of chlorosis and, of these accessions, nine (33%) allowed pustule formation. The groups representing two-row landraces and six-row barley accessions displayed varying phenotypes, containing both fully resistant and fully susceptible accessions. We extended our analysis to include two diverse panels of barley: (1) a subset of the AGOUEB panel that includes 196 domesticated, 2-row spring-type elite barley cultivars from Europe[50] and (2) the WBDC panel that encompasses 313 accessions of wild barley (*H. vulgare* subsp. *spontaneum*)[51]. Inoculation of both panels with *Pst* isolate 08/21 found that while half of accessions in the AGEOUB panel show chlorosis (98/196; 50%), no accession showed pustule formation (Supplementary Fig. 1b, c). In contrast, almost all accessions in the WBDC panel show chlorosis (301/313; 96%) and approximately one third of accessions showed pustule formation (101/313; 32%) (Supplementary Fig. 1d, e). From these results, we conclude that domestication has contributed to a differentiation in host status of barley to *Pst* isolate 08/21.

**Three major loci confer resistance to *Pst* isolate 08/21 in the cultivar Golden Promise**. The two-row spring barley cultivar Golden Promise was resistant to *Pst*, while the accession SusPtrit was fully susceptible (Supplementary Fig. 1a; Supplementary Data 1). SusPtrit was developed as a highly susceptible barley accession by selecting transgressive segregants from crosses between landraces[52]. The SusPtrit × Golden Promise doubled-haploid (DH) population has been used to investigate the genetic architecture of resistance to several non-adapted *Puccinia* spp.[45,53]. To identify loci conferring resistance to *Pst* in the cultivar Golden Promise, we inoculated the DH population with *Pst* isolate 08/21, phenotyped using macroscopic phenotypes and a complementary complimentary pair of microscopic phenotypic assays, pCOL and pPUST[47], and performed linkage analysis. Three significant loci on chromosomes 1H, 4H, and 7H contribute to resistance (Fig. 1A; Supplementary Data 2). The chromosome 7H locus corresponds to the previously characterized *Pst* resistance locus *Rps6*[54,55], whereas the loci on chromosomes 1H and 4H are near the barley powdery mildew (*Bgh*) resistance loci *Mla* (1H) and *mlo* (4H)[56,57]. We designated the loci on chromosomes 1H and 4H as *Reaction to Puccinia striiformis 7* (*Rps7*) and *Rps8*, respectively.

Collectively, these loci explain the majority of the phenotypic variation for colonization by *Pst* and loss of all three loci leads to infection (pustule formation) by *Pst*. *Rps6*, *Rps7*, and *Rps8* collectively explained 62 and 67% of phenotypic variation with

respect to chlorosis (CHL) and colonization resistance (pCOL), but only 32 and 30% of infection (INF) and pustule formation (pPUST) (Supplementary Data 2). *Rps6* and *Rps7* provide complete resistance, preventing fungal hyphal colonization and pustule formation (Fig. 1B). In contrast, *Rps8* in isolation permits a greater degree of colonization but prevents pustule formation (Fig. 1B). DH lines that lack all three loci display a fully susceptible interaction similar to SusPtrit, whereas strong resistance in Golden Promise is recapitulated in DH lines that possess all three loci. Taken together, these results demonstrate that a simple genetic architecture of three genes underlies resistance in Golden Promise: *Rps6* and *Rps7* confer resistance to early hyphal colonization, whereas *Rps8* restricts the pathogen during pustule formation.

**Rps6, Rps7, and Rps8 are present in diverse barley accessions and are principal components of resistance to *Pst* isolate 08/21.** To compare these findings with the genetic architecture of resistance in other barley germplasm, we inoculated the Foster × CIho 4196 recombinant inbred line (RIL) population and Haruna Nijo × OUH602 DH population with *Pst*. Composite interval mapping revealed that *Rps6*, *Rps7*, and *Rps8* were also prominent in these two unrelated mapping populations (Supplementary Data 2). The major component of resistance in both populations was *Rps7*, explaining 38 and 71% of the phenotypic variation for pCOL. In line with the segregation of pustule formation in the Haruna Nijo × OUH602 DH population, *Rps8* was detected at equivalent levels to the SusPtrit × Golden Promise DH population. *Rps6* was detected as a weaker QTL in the Foster × CIho 4196 population. Additional minor effect QTLs were not consistent across the different populations.

The observation of *Rps6* and *Rps7* in CIho 4196 and all three genes in Haruna Nijo suggested that these loci are more widely conserved in diverse barley germplasm. We assessed the frequency and prevalence of these genes by expanding our mapping efforts to a panel of accessions that encompasses the broader genetic diversity of barley, including wild barley. Using markers at *Rps6*, *Rps7*, and *Rps8*, we assayed the presence of these loci in 31 populations from a total of 25 diverse wheat stripe rust-resistant barley accessions crossed with susceptible accessions (Supplementary Data 2, 3, 4, and 5). *Rps6*, *Rps7*, and *Rps8* were found to varying degrees across barley diversity (Fig. 2). *Rps7* was observed in 56% (14/25) of the accessions analyzed. In five instances, *Rps7* was detected independent of *Rps6* and *Rps8* with effect sizes ranging from 10 to 95% percent variation explained (PVE) with respect to colonization resistance (Supplementary Data 2, 3, and 4). Among the wild barley accessions, WBDC172 was the only one in which *Rps7* was detected, suggesting that this gene may exist at lower frequency in wild accessions. *Rps6* was detected at a similar frequency, contributing to resistance in 44% (11/25) of the accessions. When detected, *Rps6* had large effect sizes in the majority of accessions (9/11). *Rps8* was detected in 60% (15/25) of the accessions. *Rps8* was observed in isolation from *Rps7* and *Rps6* in three accessions with effect sizes ranging from 22 to 31% PVE for colonization and 12 to 60% for pustule formation resistance. In the presence of *Rps6* and/or *Rps7*, *Rps8* consistently had smaller effect sizes due to epistasis and is likely to be present in more populations. Three two-row cultivars, Haruna Nijo, Golden Promise, and Emir, provided evidence of all three genes functioning together in the same accession. The only accession to exhibit resistance in the absence of *Rps6*, *Rps7*, and *Rps8* was Duplex, indicating that additional resistance genes exists in this accession. Taken together, these results suggest that three major loci govern resistance to *Pst* isolate 08/21 in wild, landrace, and cultivar barley accessions. The prevalence of the loci and their major contribution to resistance in all but one accession suggest that they are the principal components of resistance in barley and that reduced representation of *Rps6, Rps7,* and *Rps8* in wild barley accessions is associated with greater susceptibility to wheat stripe rust.

**Coupling of barley powdery mildew and wheat stripe rust resistance at the *Mla* locus.** Considerable genetic resources have been developed to characterize the *Mla* locus, including several panels of near-isogenic lines (NILs) that have been created using three recurrent barley cultivars: Manchuria[58], Pallas[59], and Siri[60]. While the cultivars Pallas and Siri are resistant to *Pst* isolate 08/ 21, Manchuria is susceptible. To test for colocalization of *Rps7* with diverse *Mla* alleles, we challenged paired NILs with introgressed mildew resistance genes including six *Mla* alleles (*Mla1, Mla6, Mla7, Mla10, Mla13,* and *Mla15*)[58]. The NILs harboring *Mla7* (CI 16147) and *Mla15* (CI 16153) were completely resistant to *Pst* isolate 08/21, whereas their paired mildew susceptible partners, CI 16148 and CI 16154, respectively, were completely susceptible to *Pst* isolate 08/21 (Supplementary Fig. 2). Furthermore, the *Mla1, Mla6, Mla10,* and *Mla13* haplotypes were all susceptible to *Pst* isolate 08/21. These paired differential NILs indicate close linkage between *Mla* and *Rps7*.

To resolve the association of barley powdery mildew and wheat stripe rust resistance at the *Mla* locus, we performed high-resolution recombination screens using two crosses, CIho 4196 × Morex and CI 16153 × Manchuria using the flanking markers K_963924 and K_206D11 (Fig. 3A). In the CIho 4196 × Morex recombination screen, 24 recombinants were identified from 1,136 gametes, whereas in the CI 16153 × Manchuria recombination screen, 93 recombinants were identified from 2,634 gametes. $F_3$ families derived from recombinant $F_2$ individuals in the CI 16153 × Manchuria population were inoculated with barley powdery mildew isolate CC148 (avirulent to *Mla15*) and *Mla15* was confirmed to map to the *Mla* locus. These high-resolution recombination screens confirmed *Rps7* is in coupling with *Mla* (Fig. 3B).

*Mla* is encoded by members of the *RGH1* gene family, although many haplotypes are known to contain copies of two additional NLR gene families at the locus (*RGH2* and *RGH3*)[30]. Using a dense panel of markers, we confirmed suppressed recombination over the interval containing all three NLR encoding gene families at *Mla* (Fig. 3C)[57]. Therefore, the *Mla* allele is predictive of the entire haplotype as it is inherited as a single unit. To discover the *Mla* alleles present in our diverse barley accessions, we utilized existing de novo leaf transcriptome assemblies[32,61]. Accessions CIho 4196, Betzes, Golden Promise, and Haruna Nijo harbor *Mla8*, whereas CI 16147, CI 16153, and I5 harbor *Mla7*. We confirmed previous observations that *Mla7* and *Mla15* encode the same protein[31], although we found a two amino acid differences relative to the published sequence[62]. A sequence similar to *RGH3* was found to be present using comprehensive sequence captures targeting the entire *Mla* locus of accessions Golden Promise and CI 16153. Evaluation of RNAseq data found that *RGH3* was not expressed in the accessions with available data harboring *Mla7* or *Mla8*. *RGH2* was not present in either Golden Promise or CI 16153 based on sequence capture and RNAseq. Evaluation of the *Mla8* haplotype in the recently sequenced Golden Promise genome[63] found that both *RGH2* and *RGH3* are absent from the region encompassing *Mla8*.

**Copy number variation in *Mla* alleles.** The *Mla* locus in the reference barley cultivar Morex contains a tandem duplication of 39.7 kb that contains *RGH1*, *RGH2*, and *RGH3* family members. To assess copy number variation of *Mla7* and *Mla8*, digital

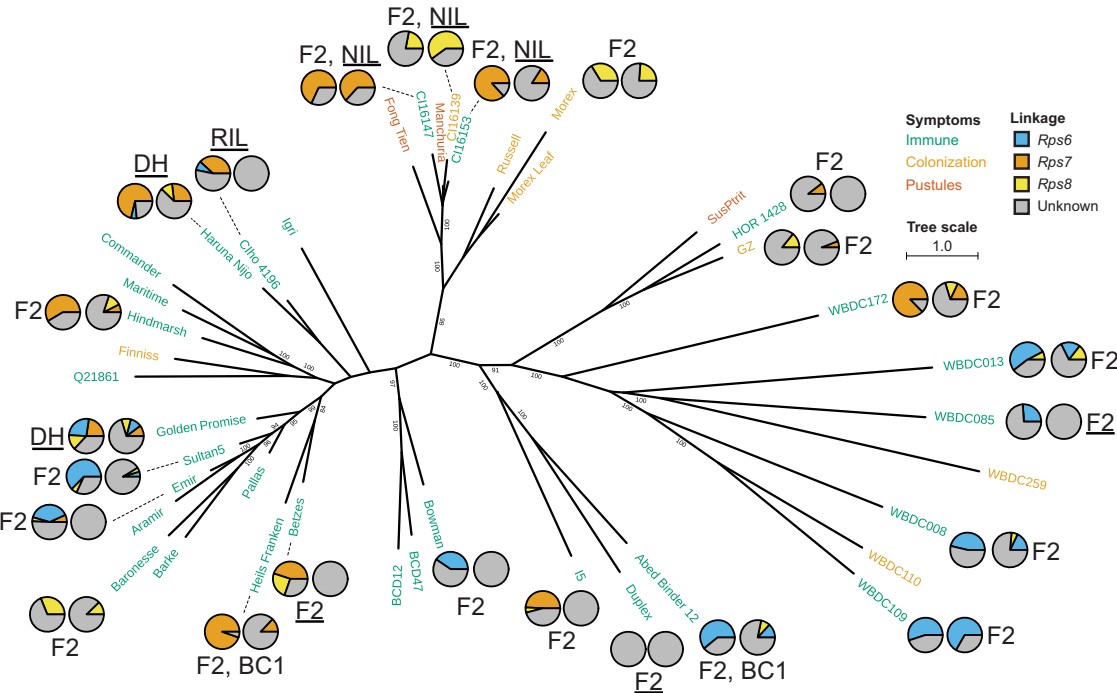

**Fig. 2 _Rps6_, _Rps7_, and _Rps8_ are present throughout diverse barley accessions.** Phylogenetic tree was generated with maximum likelihood based on SNP calling using leaf RNAseq derived from 41 genotypes. A control included RNAseq from Morex leaf. A total of 19,429 polymorphic sites were used for construction of the tree. Support over 80% is shown at branch points in the phylogeny based on 1,000 bootstraps. Scale indicates number of nucleotide substitutions. Coloring of genotypes is based on their reaction to _P. striiformis_ f. sp. _tritici_ isolate 08/21: immunity (green), colonization (yellow), and pustule formation (dark orange). Pie charts represent percent variation explained (PVE) for significant marker-trait associations (Supplementary Data 2, 3, 4). Two pie charts are shown for each population analyzed using two microscopic phenotypes, with the left and right chart representing percent colonization and percent pustule formation, respectively, with the exception of the Manchuria × Sultan 5 population, where macroscopic chlorosis and infection phenotypes were used. Mapping populations include F₂, BC₁, doubled-haploid, and recombinant inbred line populations. Underlined populations indicate comprehensive genetic maps were constructed, otherwise marker-trait associations were performed on markers near the _Rps6_, _Rps7_, and _Rps8_ loci. For accessions with multiple populations, the F₂ population was used to estimate PVE. Data not shown for HOR 2926 due to lack of RNAseq data. GZ = Grannenlose Zweizeilige. Source data are provided as a Source Data file.

droplet PCR was performed on a range of barley accessions containing _Mla8_, two independent sources of _Mla7_ (accessions Multan and Long Glumes), their corresponding Manchuria near-isogenic lines (CI 16147 and CI 16153), and CI 16155 (Manchuria NIL _Mla13_). _Mla8_ was present in a single copy across all accessions tested that harbor _Mla8_ (Supplementary Fig. 3a). The coding sequence of _Mla13_ shares 97.8% sequence identity with _Mla7_ and has only two nucleotide differences with the _RGH1_ allele in SxGP DH-47 (_RGH1.SusPtrit_). Using two independent primer pairs that amplify _Mla7_, _Mla13_, and _RGH1.SusPtrit_, we found that _Mla13_ and _RGH1.SusPtrit_ exist as single copies in CI 16155 and SxGP DH-47, respectively. In contrast, higher order copy number variation was observed in lines expressing _Mla7_, with an estimate of at least three copies (Supplementary Fig. 3b). We conclude that _Mla8_ exists as a single-copy gene, whereas the _Mla7_ and _Mla15_ haplotypes expressing _Mla7_ have multiple, identical copies.

**_Mla8_ confers resistance to barley powdery mildew and wheat stripe rust in barley**. We hypothesized that _Mla_ has dual functionality for resistance to barley powdery mildew and wheat stripe rust. To test this hypothesis, we integrated the open reading frames of _Mla1_, _Mla6_, and _Mla8_ into the native _Mla6_ genomic segment containing promoter, UTR, and terminator sequence (_p6:Mla:t6_) and transformed the barley powdery mildew and wheat stripe rust susceptible, transformable line SxGP DH-47[53] (Supplementary Fig. 4). For _Mla8_ transgenic families, advanced progeny in two independent families were identified that were

homozygous for the T-DNA insert or null. To determine whether transgenic lines carrying these _Mla_ alleles maintain resistance and race-specificity to barley powdery mildew, we inoculated transgenic lines carrying _Mla1_, _Mla6_, and _Mla8_ with 13 diverse _Bgh_ isolates that vary for their corresponding avirulence genes _AVR_{a1}_, _AVR_{a6}_, and _AVR_{a8}_. All transgenic lines harboring single copies of _Mla1_, _Mla6_, and _Mla8_ transgenes retained their specificity to recognize barley powdery mildew with corresponding _AVR_{a}_ genes based on a collection of 13 diverse isolates (Fig. 4; Supplementary Data 6). This work shows that the _Mla6_ promoter is sufficient for expression of _Mla1_, _Mla6_, and _Mla8_.

To test whether transgenic barley expressing _Mla8_ confer resistance to wheat stripe rust, we challenged progeny from primary transgenic lines of _Mla1_, _Mla6_, and homozygous _Mla8_ transgenic lines with wheat stripe rust isolate 16/035. Resistance to wheat stripe rust was only observed in transgenic lines carrying _Mla8_, whereas _Mla1_ and _Mla6_ did not confer resistance (Fig. 5A, B). Advanced transgenic families carrying single-copy inserts of _Mla8_ were challenged with wheat stripe rust isolates 08/21 and 15/151 (Fig. 6A–C,E), and in both instances, conferred resistance. When challenged with barley stripe rust isolate B01/2 the transgenic lines carrying _Mla8_ exhibited similar levels of susceptibility as controls (Fig. 6D, F). _Mla1_ and _Mla8_ are very similar (97.4%), with sequence variation only present in the LRR-encoding region. This indicates that the allelic specificity to barley powdery mildew and wheat stripe rust is determined by this 522 bp region. Therefore, we conclude that _Mla8_ has dual specificity in recognition of barley powdery mildew and wheat stripe rust.

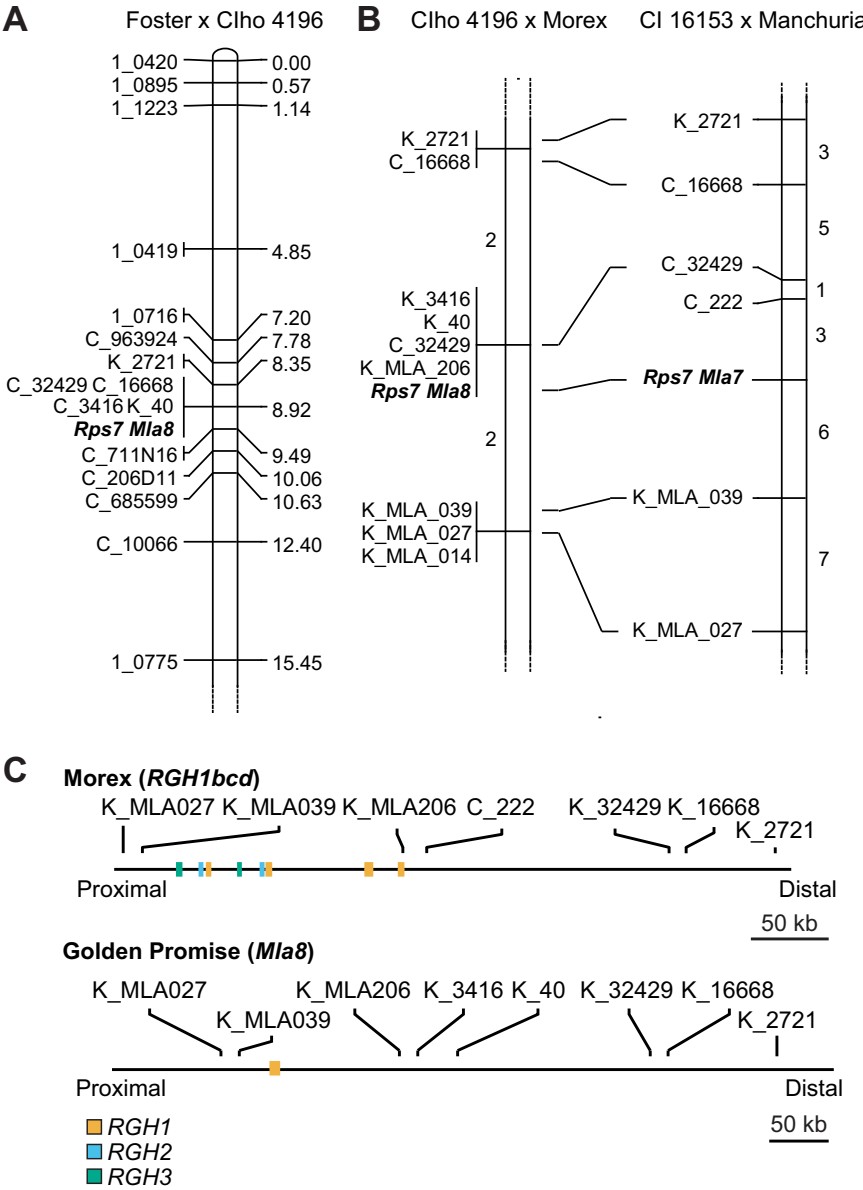

**Fig. 3 *Rps7* is in complete coupling with *Mla*. A** Genetic map of the Foster × CIho 4196 recombinant inbred line population encompassing the *Mla8/Rps7* locus. Markers K_963924/C_963924 and K_206D11/C_206D11 were used for recombination screens. Genetic distances are in cM. Prefixes designate marker type: C: Cleaved Amplified Polymorphic Sequences (CAPS) and K: Kompetitive allele-specific PCR (KASP). **B** High-resolution recombination screens were performed to resolve the relationship of *Mla* and *Rps7*. Recombination screens CI 16153 × Manchuria and CIho 4196 × Morex included 1,136 and 2,634 gametes, respectively. *Rps7* was resolved to an interval defined by K_MLA_039 and C_16668 in the CIho 4196 × Morex population and K_MLA_039 and C_222 in the CI 16153 × Manchuria population. Values between markers indicate the total number of recombinants identified. **C** Physical interval of *Mla* locus from the reference barley cultivar Morex that contains four *RGH1* genes (*RGH1a*, *RGH1bcd*, *RGH1e*, *RGH1f*) and Golden Promise that contains one *RGH1* gene (*Mla8*). In Morex, the markers K_MLA_039 and K_16668 define a physical interval of 360 kb that includes all three NLR gene families at the *Mla* locus (*RGH1* (orange), *RGH2* (blue), and *RGH3* (green)). In Golden Promise, the markers K_MLA_039 and K_16668 span a physical interval of 358 kb that includes only a single copy of *RGH1* (*Mla8*).

## Discussion

We discovered that two distinct haplotypes of *Mla* have coupled resistance to the adapted pathogen barley powdery mildew (*Bgh*; ascomycete) and non-adapted pathogen wheat stripe rust (*Pst*; basidiomycete). The majority of *Mla* haplotypes did not recognize *Pst*, suggesting that recognition resembles a similar haplotype specificity as observed for *Bgh*. Previous work demonstrated that an *Mla* ortholog in *Triticum monococcum* confers resistance to wheat powdery mildew (*B. graminis* f. sp. *tritici*) in transient assays[36], whereas *Mla* orthologs in *Aegilops tauschii* (*Sr33*) and rye (*Sr50*) confer resistance to wheat stem rust (*P. graminis* f. sp.

*tritici*)[35,37]. Interaction studies have shown direct interaction between MLA/AVR$_A$ effectors in barley powdery mildew[40,64] and Sr50/AvrSr50[39]. Taken together, the *Mla* gene family has the broad capacity to recognize effectors from a diversity of ascomycete and basidiomycete pathogens. Based on linkage with resistance to rice blast[65] and susceptibility to spot blotch[66], this recognition spectrum of the barley *Mla* locus may be even greater.

The genetic coupling of resistance to different pathogen species at NLR loci has been observed in only a few systems[67]. *Mi* from tomato confers resistance to root knot nematode (*Meloidogyne* spp.)[68], potato aphid (*Macrosiphum euphorbiae*), and whitefly

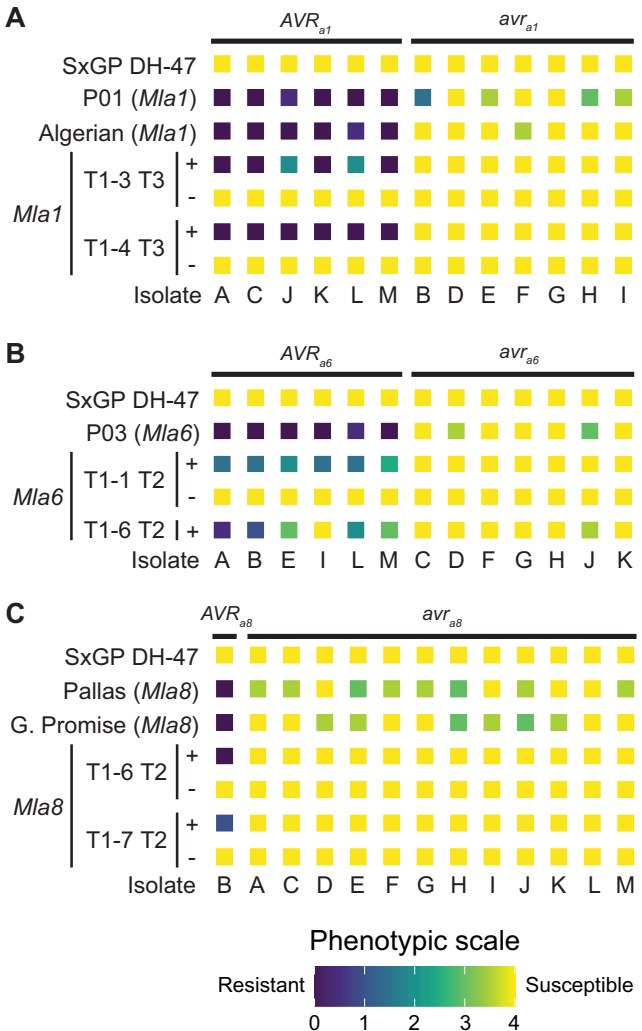

**Fig. 4 *Mla* transgenes maintain race-specific resistance to barley powdery mildew (*Blumeria graminis* f. sp. *hordei*).** Independent transformants of barley cv. SxGP DH-47 with (**A**) *Mla1*, (**B**) *Mla6*, and (**C**) *Mla8* driven by the *Mla6* promoter and *Mla6* terminator were inoculated with 13 diverse barley powdery mildew isolates. SxGP DH-47 was susceptible to all isolates. Single T-DNA insertions were sufficient to confer race-specific resistance to barley powdery mildew for *Mla1*, *Mla6*, and *Mla8*. Controls include Pallas (*Mla8*), near-isogenic lines in the Pallas genetic background: P01 (*Mla1*), P03 (*Mla6*), Algerian (*Mla1*), and Golden Promise (*Mla8*). For individual *Mla* alleles, barley powdery mildew isolates are ordered based on the presence (*AVR*) or absence (*avr*) of an avirulence gene. Isolates include 3-33 (**A**), Race I (**B**), X-4 (**C**), I-167 (**D**), K-200 (**E**), M-236 (**F**), Z-6 (**G**), C-132 (**H**), 120 (**I**), R86/1 (**J**), K-3 (**K**), KM18 (**L**), and MN-B (**M**)[86]. All experiments were performed twice with similar results and data shown is the average of these experiments. Original data is provided as Supplementary Data 6.

*Bemisia argentifolii*[26]. In *Arabidopsis thaliana*, multiple pathogen recognition at the *RPS4/RRS1* locus confers resistance to the bacterial pathogens *Pseudomonas syringae*, *Ralstonia solanacearum*, *Xanthomonas campestris*, and the fungal pathogen *Colletotrichum higginsianum*[24,69]. NLR recognition of taxonomically diverse pathogens that do not share a common effector repertoire suggests that either (1) effectors from these pathogens contain a conserved structure that directly interacts with the NLR, (2) the NLR has the capacity to recognize different effectors through different binding sites, or (3) the NLR guards a host protein that is a shared effector target by multiple pathogens. In the latter

model of recognition, these effectors may or may not be structurally related. In *A. thaliana*, recognition of *P. syringae* and *R. solanacearum* by the NLR pair RPS4 and RRS1 requires the WRKY domain of RRS1, and requires the effectors AvrRps4 and PopP2, respectively[70,71]. While the corresponding effector from *C. higginsianum* is not known, AvrRps4 and PopP2 are sequence unrelated, suggesting that the WRKY domain is a conserved target by multiple pathogens. An initial analysis of the *Pst* genome[31] did not identify a homolog of cloned AVR$_a$ proteins[38,40,64] of barley powdery mildew. In the present study, we favor the first and second models of recognition, based on previous work that has found that the majority of MLA proteins likely bind their corresponding AVR$_a$ directly[40,64]. *Mla1* is functional against barley powdery mildew when transferred to *Arabidopsis thaliana*, which further supports a model of direct interaction with an effector[72]. Lastly, it is possible that different alleles of MLA may have differing modes of recognition, as MLA9 unlike other tested MLA proteins, was found to not elicit an HR when heterologously expressed with its corresponding effector in *Nicotiana benthamiana*[40]. Identification of the recognized effector in wheat stripe rust will be essential to establish the mode of recognition.

Genetic coupling at *Mla* demonstrates that selection for resistance to one pathogen can directly influence resistance to other pathogens, including non-adapted pathogens. The implications of multiple pathogen recognition suggest that selection for one pathogen could lead to greater resistance (mutualistic) or susceptibility (antagonistic) to another pathogen. *Mla* is a target of breeding, often through the introduction of exotic alleles from wild barley[33,73]. As we observed considerable susceptibility in wild barley, it suggests that the majority of *Mla* alleles present in these accessions do not confer resistance to wheat stripe rust (Supplementary Data 7). More broadly, marker-assisted selection and the fixation of resistance genes in plant breeding programs are routine[74] and there is a risk associated with not understanding the broader capacity of resistance genes in maintaining a barrier to host range dynamics. As plant breeders seek to incorporate novel sources of resistance from primary, secondary, and tertiary gene pools of crop species, they also expose agricultural systems to the introduction of susceptibility genes or the removal of genes that maintain barriers to non-adapted pathogens. As marker-assisted selection is often used for the incorporation of exotic alleles, a novel form of susceptibility may not be detected until after the release of a cultivar, as was observed for the host jump of Victoria blight (*Bipolaris victoriae*) onto oats containing the oat crown rust resistance gene *Pc2*[75] and the host range expansion of blast (*Pyricularia oryzae*) through the wheat cultivar Anahuac, which lacked a crucial resistance gene[76]. The implication of multiple pathogen recognition in host species specificity highlights the potential impact of current plant breeding practices to host range expansions.

Barley is grown in temperate regions of the world and occupies an agricultural niche similar to wheat[77]. Thus, barley, inevitably, is exposed to great inoculation potential of wheat-adapted pathogens. Despite this, barley remains highly resistant to non-adapted *Pst* yet is readily infected by the adapted form, *Psh*[78,79]. A critical example of the durability of resistance is found in Australia, where neither wheat stripe rust nor barley stripe rust were present until 1979 when *Pst* was introduced[78]. Despite high pathogen pressure, *Pst* has not made a host jump onto barley. The introduced *Pst* in Australia is genetically similar to *Pst* isolate 08/21[80,81]. Several barley accessions used in this work, including Commander, Maritime, Hindmarsh, and Finniss are Australian cultivars, and Hindmarsh was found to carry *Rps7* (*Mla8*) and *Rps8*. This raises the question, how has barley maintained resistance to *Pst* in Australia despite significant infection pressure in

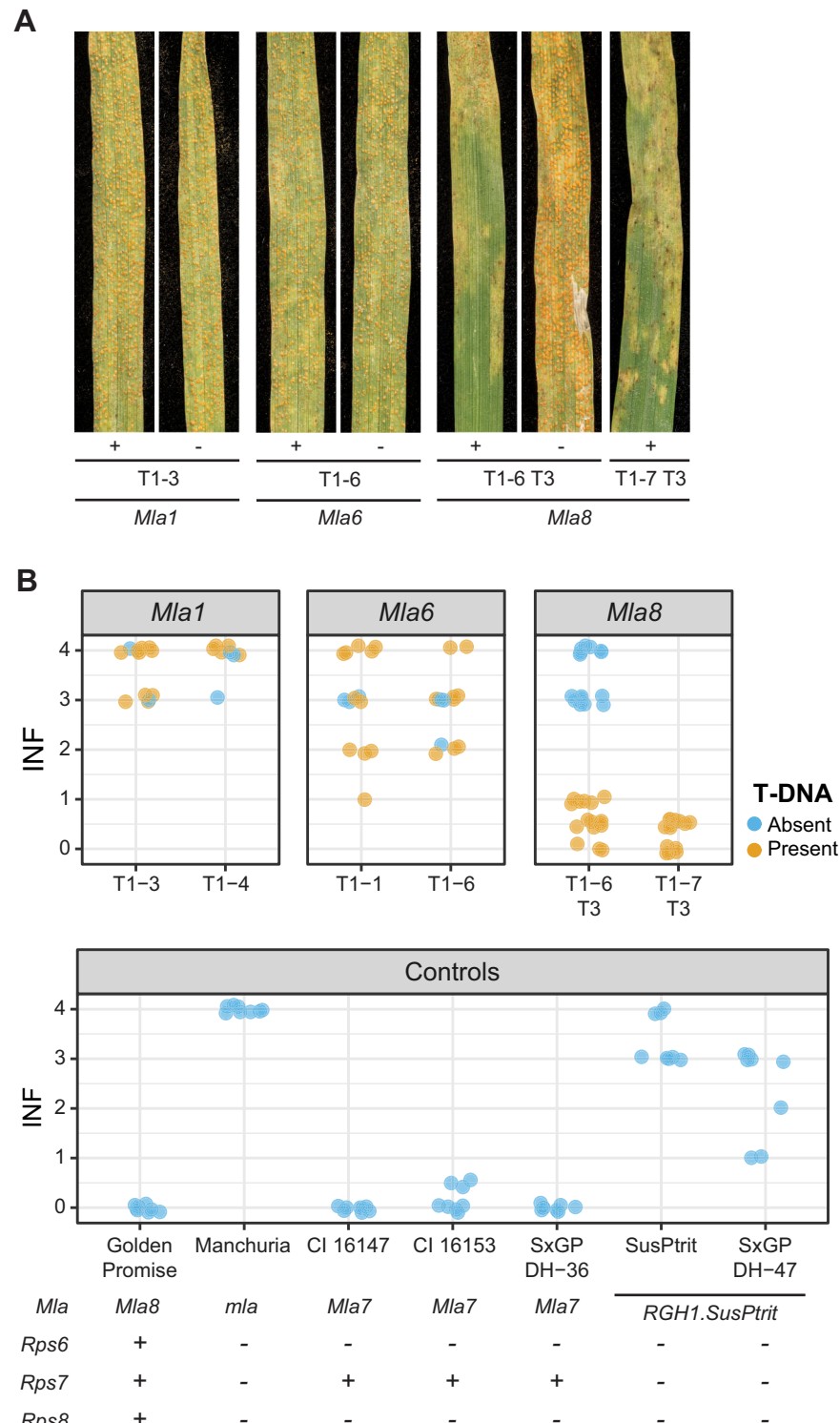

**Fig. 5 Allele-specific resistance of *Mla* to wheat stripe rust (*P. striiformis* f. sp. *tritici*) isolate 16/035.** Wheat stripe rust susceptible barley cv. SxGP DH-47 was transformed with *Mla1*, *Mla6*, and *Mla8* driven by the *Mla6* promoter and *Mla6* terminator. Two single-copy insert lines were identified for *Mla1* (T1-3 and T1-4) and *Mla6* (T1-1 and T1-6). For *Mla8*, T3 homozygous progeny present (T1-6 and T1-7) and absent (T1-6) for the transgene were used. Resistance to wheat stripe rust isolate 16/035 was observed in transgenic barley lines with single copy of *Mla8* and in the controls Golden Promise (*Rps6*; *Mla8/Rps7*; *Rps8*), CI 16147 (*Mla7/Rps7*), CI 16153 (*Mla7/Rps7*), and SxGP DH-36 (*Mla8/Rps7*). Susceptibility to *P. striiformis* f. sp. *tritici* isolate 16/035 was observed in transgenic barley lines with T-DNA containing *Mla1* and *Mla6*. **A** Representative leaf phenotypes in the presence (+) or absence (−) of T-DNA for individual *Mla* alleles. **B** Infection phenotypes for individual leaves in T1 families and controls. Presence or absence of T-DNA is shown in orange and blue, respectively. All phenotypes are on a scale from 0.0 to 4.0 in 0.5 increments represent the degree of leaf area covered with pustules. Transparency and jittering were used to visualize multiple overlapping data points. An extended set of controls for *P. striiformis* f. sp. *tritici* isolate 16/035 can be found in Supplementary Fig. 5 and 6. Source data are provided as a Source Data file.

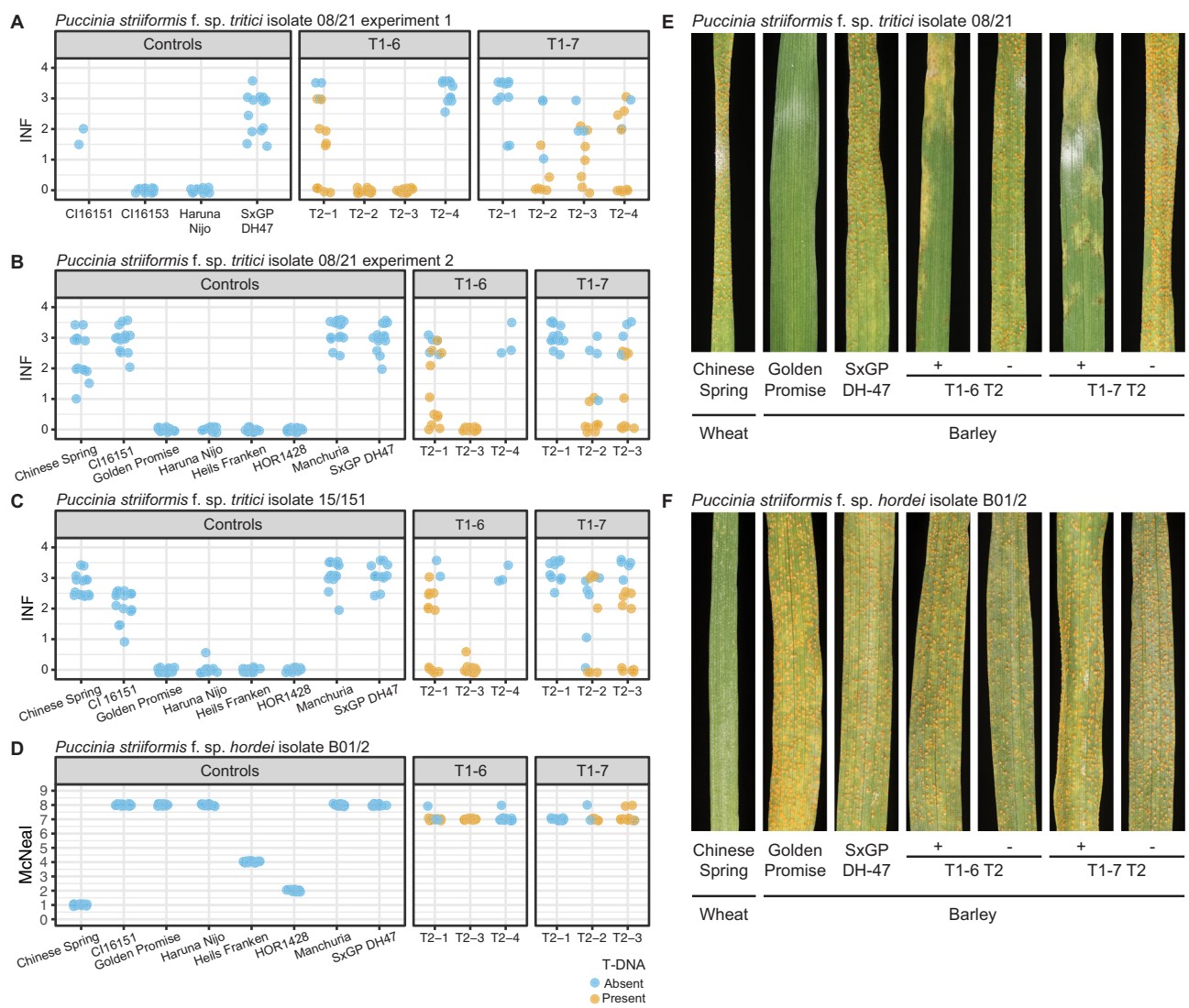

**Fig. 6 *Mla8* confers resistance to *P. striiformis* f. sp. *tritici* isolates 08/21 and 15/151, but not *P. striiformis* f. sp. *hordei* isolate B01/2.** Advanced T$_2$ progeny from two independent single insert *p6:Mla8:t6* transgenic families in the SxGP DH-47 genetic background were inoculated with *P. striiformis* f. sp. *tritici* isolates (**A**, **B**) 08/21 and (**C**) 15/151, and (**D**) *P. striiformis* f. sp. *hordei* isolate B01/2. Representative photos are shown for (**E**) *P. striiformis* f. sp. *tritici* isolate 08/21 and (**F**) *P. striiformis* f. sp. *hordei* isolate B01/2. Phenotypic scales used include infection (INF) for wheat stripe rust[47] (0.0 to 4.0 in 0.5 increments represent the degree of leaf area covered with pustules) and the McNeal scale for barley stripe rust[90]. Presence or absence of the *p6:Mla8:t6* transgene is color coded orange or blue, respectively. Transparency and jittering were used to visualize multiple overlapping data points. Progeny from transgenic family T1-6 were selfed, generating hemizygous T-DNA families (T1-6 T2-2 and T1-6 T2-3), homozygous null (T1-6 T2-1). Progeny from transgenic family T1-7 were selfed, generating homozygous T-DNA (T1-7 T2-2), hemizygous T-DNA (T1-7 T2-1), homozygous null (T1-7 T2-3). Controls include the wheat accession Chinese Spring, and barley accessions CI16151 (*rps6 rps7*(=*Mla6*) *rps8*), Golden Promise (*Rps6 Rps7*(=*Mla8*) *Rps8*), Haruna Nijo (*Rps6 Rps7*(=*Mla8*) *Rps8*), Heils Franken (*rps6 rps7*(=*Mla37*) *rps8*), and HOR1428 (*rps6 Rps7 rps8*), Manchuria (*rps6 rps7 rps8*) and SxGP DH-47 (*rps6 rps7 rps8*). Heils Franken and HOR 1428 are resistant controls to *P. striiformis* f. sp. *hordei* isolate B01/2. Source data are provided as a Source Data file.

the field? We hypothesize that three factors contribute to maintaining resistance. First, the majority of accessions carry multiple resistance genes against *Pst*. The presence of multiple genes likely contributes to their durability. Second, while *Rps6*, *Rps7*, and *Rps8* were identified across both domesticated and wild barley, we also identified several minor effect QTL that varied between accessions. These loci represent a broad complex of major and minor effect resistance genes that contribute to the non-adapted status of barley. Lastly, the underlying genetic architecture that contributes to maintaining host species specificity to *P. striiformis* may change based on the developmental stage of the plant. Niks and colleagues (2013) demonstrated that barley accessions susceptible to two non-adapted *P. striiformis* at the seedling stage had variable response at the adult stage[82]. As only four accessions

were tested with *P. striiformis* f. sp. *bromi* and one accession with *Pst*, additional studies are required to assess the contribution of developmentally regulated resistance in maintaining resistance to non-adapted *P. striiformis*. Further work will be required to establish the broader role of *Rps6*, *Rps7*, and *Rps8* in the context of natural variation of barley and stripe rust and how the developmental stage may influence the underlying genetic architecture of immunity in barley to non-adapted stripe rust.

The coupling of resistance directly addresses a long-standing question in nonhost resistance: How can selection act on a phenotype that does not have a direct fitness benefit? If the plant immune system has the broader capacity to recognize diverse pathogen species, then selection against adapted pathogens may maintain resistance to non-adapted pathogens. Two alternative

non-exclusive models may explain this phenomenon. First, pathogens may be able to exploit only a limited number of host targets for suppressing immunity, many of which are guarded by NLRs. Alternatively, the basis of effector function may require the adoption of a conserved structure, which is directly recognized by NLRs. With respect to the evolution of immunity, population genetic models previously focused on selection acting on single dominant *R* genes in host systems, which lead to either recurrent purifying selection or balancing selection[83]. The coupling of resistance to multiple pathogens indicates that more complex models will be required to describe the selective forces driving the evolution of immunity in plants.

## Methods

**Plant and fungal materials**. Barley accessions were obtained from diverse sources that are referenced in Supplementary Data 1. The Foster × CIho 4196 recombinant inbred line population ($N = 89$) was provided by A. Kleinhofs (Washington State University, WA, USA)[84]. The Haruna Nijo × OUH602 doubled-haploid population ($N = 94$) was provided by Kaz Sato (Okayama University, Okayama, Japan)[85]. The SusPtrit × Golden Promise doubled-haploid population was provided by Rients Niks (Wageningen University, Netherlands)[53]. All plants were subjected to single seed descent. *Pst* isolates 08/21, 15/151, and 16/035 were collected in 2008, 2015, and 2016, respectively, in the United Kingdom and maintained at the National Institute of Agricultural Botany (NIAB). *Psh* isolate B01/2 was collected in 2001 in the United Kingdom and maintained at NIAB. *Pst* and *Psh* isolates were increased on susceptible wheat and barley cultivars, respectively, collected, and stored at 6 °C. *B. graminis* f. sp. *hordei* isolate CC148 was collected in the United Kingdom and propagated on barley cv. Manchuria (CI 2330). A subset of 13 isolates of *B. graminis* f. sp. *hordei* was selected from a gene bank of the pathogen containing 59 reference isolates collected in all nonpolar continents over a period of 66 years (1953–2019) and kept at the Agricultural Research Institute Kroměříž Ltd. Virulence patterns to 35 standard barley genotypes are shown in Dreiseitl (2019)[86]. Before inoculation all isolates were checked for their purity and their correct virulence phenotypes were verified on standard barley lines[59]. The isolates were multiplied on leaf segments of a susceptible cultivar Bowman.

**Genomic DNA isolation**. Genomic DNA from all populations was extracted from leaf tissue following a 96-well PCR-grade genomic DNA isolation procedure for cereal leaf tissue modified from Stewart and Via[87]. Second leaves were placed into a 96-well plate (Abgene, #AB-0661) prefilled with three 5/32" stainless steel balls (OPM Diagnostics, #SP-2150). Plates were sealed with a permeable membrane (Excel Scientific, #B-100) and placed into a lyophyllizer (Thermo Scientific, PowerDry LL3000) for 48 h. After removal, plates were sealed with silicon capmats and placed in a TissueLyser I bead mill (Qiagen) for 1 min at 1500 rpm, then reversing plate orientation and repeating the grind. Plates were spun for 2 min at $2500 \times g$ to remove residual powder from capmats. A CTAB extraction buffer was prepared as follows for a single plate: 0.35 g of CTAB, 25.5 mL of ddH$_2$0, 3.5 mL of 1.0 M Tris at pH 8.0, 4.9 mL of 5.0 M NaCl, 700 µL of 0.5 M EDTA at pH 8.0, and 350 uL of 14 M 2-Mercaptoethanol. All ingredients were added in the described order, with BME added prior to genomic DNA extraction. Next, 300 µL of CTAB extraction buffer was applied to each well. Plates were loaded into custom made wooden clamps and mixed several times. The entire assembly was incubated for 45 min at 65 °C, during which the assembly was mixed by inversion every 15 min. Plates were spun for 1 min at $2500 \times g$, 100 µL of 5.0 M KOAc was added to each well, resealed and mixed by inversion, and placed into an ice bath for 20 min. Plates were again spun for 1 min at $2500 \times g$, and 150 µL of chloroform:isoamyl alchohol (v/v, 24:1) was added to each well, resealed, placed in clamps, and mixed by inversion for 5 min. Plates were then spun for 10 min at $2500 \times g$ at room temperature. Approximately 200 µL of the upper aqueous layer was applied to 120 µL of isopropanol, resealed, inverted 20 times, and then spun for 15 min at $2,500 \times g$. After centrifugation, the supernatant was decanted and the plate was blotted dry. The pellet was resuspended in 200 µL TE with 0.2 mg/mL of RNAse (Roche), inverted several times, and incubated for 10 min at 65 °C. After spinning the plate, 300 µL of 7:1 isopropanol:4.4 M NH$_4$Ac was added, the plate inverted several times, and then spun for 10 min at $2500 \times g$. After centrifugation, the supernatant was decanted and the plate was blotted dry. Pellets were rinsed with 250 µL of 70% ethanol, spun for 3 min at $2500 \times g$, decanted and blotted dry, then placed in a 65 °C oven for 10 min to remove residual ethanol. The genomic DNA pellet was resuspended in 100 µL of PCR-grade TE.

**Pathogen assays**. Pathogen assays with *P. striiformis* were performed using a suspension of urediniospores talcum powder, at a weighted ratio of spores:talcum powder of 1:16, and applied to leaves using compressed on a spinning table. Immediately after inoculation, plants were sealed in a high-density polyethylene bag and stored at 8 °C for 48 h. The sealed environment was used to ensure high humidity. Plants were grown in the same controlled environment room before and after inoculation with the following conditions: 16 h light/8 h dark), with day

temperature of 18 °C and night temperature of 11 °C. Lighting was supplied by halogen light bulbs. Experimental design for mapping population involved four groups of eight seeds planted in different equidistant quadrants in a 1 L pots using a peat-based compost. Seedlings were were macroscopically phenotyped 14 days post-inoculation and the first leaf sampled for microscopic phenotyping.

Pathogen assays using the diverse collection of *Bgh* isolates, inoculation and evaluation was previously described[88]. Briefly, seeds of transgenic families were planted in gardening peat substrate in an 80 mm diameter pot (80 mm diameter) and grown in a mildew-proof greenhouse under natural daylight. After 14 days, 15 mm leaf segments were taken from the center of fully expanded first leaves (second leaf just emerging). For each transgenic family and controls, three segments were arrayed along with four segments of the susceptible cultivar Bowman oriented diagonally. Leaves were placed with adaxial surfaces facing up in a 150 mm Petri dish on water agar (0.8%) containing benzimidazole (40 mg/L), a leaf senescence inhibitor. Inoculation was performed by placing leaf segments at the bottom of a cylindrical metal settling tower (150 mm diameter, 415 mm height). Conidiaspores from fully developed pathogen colonies grown on cultivar Bowman were harvested by manual shaking onto a square piece (40 × 40 mm) of black paper to visually estimate the amount of inoculum. A blowpipe was formed by rolling the paper, and blown through a side hole of 13 mm diameter in the upper part of the settling tower over the Petri dish at a concentration of approximately 8 conidia/mm². Dishes with inoculated leaf segments were placed in a controlled environment room with 20 ± 2 °C under artificial light (cool-white fluorescent lamps providing 12 h light at 30 ± 5 µmol m$^{-2}$ s$^{-1}$). Infection responses (IR; virulent/avirulent) were scored at seven days after inoculation on the central part of leaf segments on a scale 0–4, where 0 is equivalent to no visible mycelium or sporulation, and 4 represents strong mycelial growth and sporulation[89]. Scoring was repeated one day later, and significant differences were rescored. A total of two replications were performed for all lines. A set of 13 reaction types provide an infection response array (IRA) for each line and hypothetical resistance gene specificity in tested lines were postulated by comparing their IRAs with known IRAs of barley genotypes possessing known resistance genes.

**Macroscopic phenotyping**. For *Pst*, macroscopic symptoms were evaluated on the first leaf of all seedlings at 14 days post-inoculation based on chlorosis and infection (presence of fungal pustules), both scored on a scale of 0 to 4, with increments of 0.5. In both cases, the phenotypic scale reflects the surface area expressing the phenotype. A score of 0 was symptomatic of no expression of the phenotype, i.e. no chlorosis or no pustules, and a score of 4 was indicated full expression (i.e. 100% of the surface area). For *Pst*, macroscopic symptoms were evaluated on the first leaf of all seedlings at 14 days post-inoculation using the McNeal scale[90]. For *B. graminis* f. sp. *hordei*, macroscopic symptoms were evaluated on the first leaf of all seedlings at 7 days post-inoculation based on the surface area of leaf covered by colonies, scored on a scale of 0–4, with increments of 0.5. Equivalent to scoring for *Pst*, a score of 0 was symptomatic of no expression of the phenotype, i.e. no pustules observed, and a score of 4 was indicated full expression (*i.e.* 100% of the surface area covered in pustules).

**Microscopic phenotyping**. We adapted a protocol described by Ayliffe et al.[91] that uses a stain that binds N-acetyl-glucosamine (a major component of chitin) for the visualization of intercellular fungal growth. A detailed description of this protocol is made in Dawson et al.[47]. Briefly, leaves were harvest and placed in 1.0 M KOH (5 mL of KOH per barley leaf) with a droplet of surfactant (Silwet L-77, Loveland Industries Ltd.). The leaf tissue was incubated in KOH solution at 37 °C between 12 and 16 h and then washed three times in 50 mM Tris pH 7.5. After decanting of the wash solution, a 1.0 mL stain solution consisting of 20 µg/mL wheat germ agglutinin conjugated with FITC (L4895-10MG; Sigma-Aldrich) in 50 mM Tris pH 7.5 was applied to leaf tissue and incubated overnight. Leaf tissue was then washed with water, mounted, and observed under blue light excitation using a mercury lamp with GFP filter in an AxioPhot fluorescence microscope (Zeiss) to visualize WGA-FITC-stained *Pst*. We developed a microscopy-based phenotypic assay to evaluate stripe rust reaction as percent colonization (pCOL), this was achieved by evaluating disjoint fields of view (FOV) covering the surface area of the leaf. Within each FOV, colonization of *Pst* was determined to be <15%, 15–50% or >50% and given scores of 0, 0.5, or 1, respectively. The final pCOL score was determined by averaging these scores based on an entire leaf.

**Genotyping**. CAPS and SSLP markers were generated based on several known SNP-based markers (Supplementary Data 8) or existing report[92]. Markers were developed using barley oligo pooled assay markers[48,93], SNPs derived from sequence variation in the *Mla* contig from barley cv. Morex[30,57], and from previously characterized genotype by sequencing markers[94,95]. KASP genotyping was performed at the John Innes Centre genotyping facility (Supplementary Data 9). Oligonucleotide assay (OPA) and Sequenom genotyping was carried out as described by Dawson et al.[55]. OPA genotyping was performed at the University of California, Los Angeles Southern California Genotyping Consortium (Los Angeles, CA, USA) using 1,536 SNP-based markers (BOPA1)[48]. All Sequenom genotyping was performed by the Iowa State University Genomic Technologies Facility (Ames, IA, USA). CAPS marker reactions were performed using 2 µL buffer (10×), 0.4 µL

dNTPs, 0.4 μL forward primer, 0.4 μL reverse primer, 0.2 μL *Taq* polymerase, 2 μL gDNA at 10 ng/μL, and 14.6 μL water. PCR cycles involved initial denaturation step at 94 °C for 5 min, then 35 cycles of 94 °C for 20 s, annealing at 56 °C for 30 s and primer extension at 72 °C for 1 min. A final extension at 72 °C for 5 min before being held at 16 °C. Manufacturer's instructions were used for all restriction enzyme digestions. Electrophoresis was performed using 2.0 % TBE agarose gels stained with ethidium bromide. Imaging was performed using a Bio-Rad ChemDoc XRS + imaging system and markers were visually assessed. For Sequenom marker development, SNP sequences were extracted in IUPAC format with 40–60 bp flanking sequence (Supplementary Data 10) and used as a template for primer design using MassARRAY software v3.1 for the multiplexing up to 32 SNP assays. Genetic maps were constructed based on the barley consensus genetic map[93] and validated using R/qtl recombination fraction plots[96].

**QTL and marker-trait association analysis**. For populations with comprehensive genetic maps, composite interval mapping was performed using QTL Cartographer version 1.17e[97,98] using a walking speed of 2 cM, a window size of 10 cM, and a maximum of five background parameters. Background parameters were selected using the FB method with default settings ($p(F_{in}) = p(F_{out}) = 0.1$). For each trait, the experiment-wise threshold (EWT) was determined using 1000 permutations with reselection of background parameters[99]. We automated QTL analysis and figure generation in the QKcartographer suite of Python scripts that are maintained on GitHub (https://github.com/matthewmoscou/QKcartographer). For populations with markers near *Rps6*, *Rps7*, and *Rps8*, initial marker-trait regression was performed using a three additive QTL model including all loci. The model was then reduced based on a minimum threshold of $p = 0.05$. Marker-trait association were evaluated for significance using 1000 permutations. Percent of variation explained (PVE) was estimated using R/qtl fitqtl with all significant QTLs under an additive model.

**Recombination screens**. Two recombination screens were performed using F₂ progeny derived from the crosses CIho 4196 × Morex and CI 16153 × Manchuria. In both screens, recombinant individuals were selected using the flanking markers K_963924 and K_206D11. Informative markers for each population were applied to all recombinants derived from the recombination screens. A minimum of sixteen individuals from F₂:₃ families were independently assessed using *Pst* isolate 08/21 for both populations and *B. graminis* f. sp. *hordei* isolate CC148 for the CI 16153 × Manchuria F₂:₃ families.

**Transcriptome sequencing and assembly**. Transcriptome sequencing and assembly was carried out according to Brabham et al.[32]. Briefly, total RNA was extracted using a Trizol-phenol based protocol according to manufacturer's protocol (Sigma-Aldrich; T9424). Barcoded Illumina TruSeq RNA HT libraries were constructed and pooled with four samples per lane on a single HiSeq 2500 lane run in Rapid Run mode. Sequencing was performed using 150 bp paired-end reads. Paired-end reads were assessed for quality using FastQC and trimmed before assembly using Trimmomatic (v0.32) with parameters set at ILLUMINA-CLIP:2:30:10, LEADING:3, TRAILING:3, SLIDINGWINDOW:4:15, and MIN-LEN:100. These parameters were used to remove all reads with adapter sequence, ambiguous bases, or a substantial reduction in read quality. De novo transcriptome assemblies were generated using Trinity with default parameters (version 2013-11-10)[100]. Alignments to RNAseq assemblies and controls were performed using bowtie2 (version 2.3.4.1) using default parameters.

**Sequence capture and PacBio SMRT sequencing**. Sequence capture and PacBio SMRT sequencing of NLR encoding genes (RenSeq) were carried out according to Witek et al.[101]. Briefly, gDNA extracted using a CTAB method, normalized to 3 μg, and sheared with a Covaris S2 sonicator (settings Duty Cycle 20%, Intensity 1, Cycle Burst 1000, Time 600 s, Sample volume 200 μL) to an average length of 3 to 4 kb. Sequencing libraries were prepared using the NEBNext Ultra DNA Library Prep Kit for Illumina (NEB, MA, USA) using DNA fragments greater than 2 kb using Agencourt AMPure XP beads. Capturing NLR gene fragments was performed using a custom MYcroarray MYbaits bait library based on the barley resistance gene space (TSLMMHV1).

After ligation of Illumina sequencing adapters, the sample was purified with AMPure XP beads, and the subjected to eight cycles of PCR amplification using indexed PCR primers (NEBNext Multiplex Oligos for Illumina, New England Biolabs) and the Illumina PE1.0 PCR primer. The Bioanalyzer DNA 1000 chip (Agilent) was used for quality assays and the average fragment sizes and concentrations determined with a Qubit dsDNA assay. Approximately 500 ng of the prepped library was hybridized in hybridization buffer (10x SSPE, 10X Denhardt's solution, 10 mM EDTA, 0.2% SDS) to the biotinylated RNA baits for 20 h at 65 °C on a thermocycler. Recovery of DNA using magnetic streptavidin-coated beads was performed by adding 30 μL Dynabeads MyOne Streptavidin C1 (Invitrogen, Life Technologies) that had been washed three times and resuspended in binding buffer (1 M NaCl; 10 mM Tris-HCl, pH 7.5; 1 mM EDTA). After 30 min at 65 °C, beads were pulled down and washed three times at 65 °C for 10 min with 0.02% SSC/0.1% SDS followed by resuspension in 30 μL of nuclease-free water. PCR amplification using Kapa HiFi HotStart Ready Mix (Kapa Biosystems) and Illumina P5 and P7 primers was performed with 26 cycles. Size fractionation of the amplified library was performed using a Sage Scientific Electrophoretic Lateral Fractionator (SageELF, Sage Science) using a 0.75% SageELF agarose gel cassette. The fractions that ranged between 3 and 4 kb were pooled and purified with AMPure PB beads (Pacific Biosciences). SMRTbell Template Prep Kit 1.0 (Pacific Biosciences) was used to assemble the library according the 2-kb Template Preparation and Sequencing protocol (www.pacificbiosciences.com/support/pubmap/documentation.html). PacBio RSII sequencing using C4-P6 chemistry was performed at the Earlham Institute (Norwich, UK), using four SMRT cells for each barley accession.

The complete design process of the library is described in Brabham et al.[32]. Briefly, the capture design TSLMMHV1 includes 99,421 100 mer baits that target the barley NLR gene space based on available genomic resources as well as the entire *Mla* interval, excluding repetitive sequence. This includes allelic variation in *RGH1*, *RGH2*, and *RGH3* gene families. In general, a target of 2x non-redundant coverage was achieved for all NLRs identified in the genomes of the barley accessions Barke, Bowman, and Morex, the full length cDNA from barley accession Haruna Nijo, and transcriptomes of barley accessions Abed Binder 12, Baronesse, CI 16153, CIho 4196, Manchuria, Pallas, Russell, and SusPtrit.

Preprocessing of PacBio circular consensus reads involved selecting only those reads with three or more passes, trimming of reads included the first first and last 70 bp, and size selected to reads less than 4 kb. De novo assembly of PacBio circular consensus sequences was performed using Geneious (v10.2.3) using custom sensitivity parameters for assembly: don't merge variants with coverage over approximately 6, merge homopolymer variants, allow gaps up to a maximum of 15% gaps per read, word length of 14, minimum overlap of 250 bp, ignore words repeated more than 200 times, 5% maximum mismatches per read, maximum gap size of 2, minimum overlap identity of 90%, index word length 12, reanalyze threshold of 8, and maximum ambiguity of 4.

**Phylogenetic tree construction**. The barley reference gene space[102] was used as template for performing alignments with bwa mem (version 0.7.5a-r405) using default parameters. Samtools (version 0.1.19-96b5f2294a) was used for file conversion and application of the requirement that reads mapped in a proper pair (-f2). Reads were sorted and duplicate reads removed. Coverage of reads was determined using bedtools (version v2.17.0). SNPs and InDels were called using VarScan (version 2.3.8) with default parameters. The QKgenome suite (version 1.1.1) was used to assess allelic diversity in barley coding sequence among diverse genotypes[80]. The QKgenome_conversion.py script was used to assess nucleotide variation with read depth of equal to or greater than 20 across the entire coding sequence[80]. A threshold of at least 90% was required for SNPs and InDels. All genes harboring InDels or mutations that disrupted the coding sequence (truncations) were excluded in the analysis. Multiple sequence alignment of polymorphic sites was performed using the QKgenome_phylogeny.py script (https://github.com/matthewmoscou/QKphylogeny). Construction of the phylogenetic tree was performed with RAxML (version 8.2.9) using the GTRGAMMA nucleotide model and rapid hill-climbing mode. A total of 500 bootstraps were performed and sufficient based on the bootstrap convergence test.

**Construct development and plant transformation**. The open reading frames of *Mla8* and *Mla6*, and the promoter, UTRs, and terminator regions of *Mla6* were amplified from barley accessions Haruna Nijo (*Mla8*) and CI 16151 (*Mla6*)[103], respectively, using GoTaq Long PCR Master Mix (Promega). PCR bands were gel purified and cloned into pGem-T Easy vector (Promega) using 3:1 insert:vector molar ratios. Plasmid DNA from colonies were isolated and inserts were verified by Sanger sequencing. *Mla1* is identical to *Mla8* except for a 3′ region of 495 bp (2,383 to 2,877 bp). A template of 759 bp fragment encompassing this region was synthesized by IDT based on the genomic sequence of *Mla1* (NCBI AY009938 [https://www.ncbi.nlm.nih.gov/nuccore/AY009938.2/]). Assembly of the correct fragments into pBract202 binary vector (BRACT) was performed by Gibson reaction[104]. PCR fragments bearing 20 bp overlapping to the flanking region were produced with Phusion High-Fidelity DNA Polymerase (NEB) and 40-mer primers (Supplementary Data 11). After gel purification fragments were combined in an equimolar ratio, with a total DNA input of 200 ng in a 5 μL total volume. The mixture was added to 15 μL of Gibson assembly master reaction and incubated for 1 hr at 50 °C. After incubation, 10 μL of the assembly mix were transformed into chemically competent DH5α *E. coli* cells. The Gibson assembly master reaction was prepared by mixing 699 μL water, 320 μL 5x ISO buffer (500 mM Tris-HCl, pH 7.5, 250 mg/mL PEG-8000, 50 mM MgCl₂, 50 mM DTT, 1 mM dNTPs, 5 mM NAD), 0.64 μL T5 exonuclease (NEB, 10 U/μL), 20 μL Phusion DNA polymerase (NEB, 2U/μL) and 160 μL *Taq* DNA ligase (NEB, 40 U/μL). Plasmid DNA was isolated from colonies and assessed by restriction digestion to verify the assembly of all the fragments. Integrity and sequence of positive clones were confirmed using Sanger sequencing. Transformation of the wheat stripe rust susceptible SusPtrit × Golden Promise DH-47 from the SxGP doubled-haploid population[53] was performed by using the technique described by Hensel et al.[105] using the hygromycin resistance gene (*hyg*) as a selectable marker.

**Copy number variation**. Fresh leaves of plants were harvested and freeze-dried for 48 h prior to genomic DNA extraction using a CTAB-based protocol. DNA

concentrations were measured using Qubit dsDNA assay. Droplet-Digital PCR (ddPCR) was used for detection of copy number. Primers for ddPCR include *Actin* (AK365182) for normalization: forward 5′-TTTGGTGCTAGCGTGGGG-3′ and reverse 5′-AGCAAGTACTAGGGGCCGA-3′, *Mla8*: forward 5′-GCCGTGTATAG CAGAAGGTG-3′ and reverse 5′-CTGCTACATACCTCGAGAGCA-3′, and *Mla7* forward 5′-TCTGGCCATGCAGAAGCC-3′ reverse 5′-CCAGACACGAGCAACC GT-3′. The *Mla7* primers are identical to *Mla13* (CI 161555) and *RGH1* in SusPtrit and were used for amplification of these genes. PCR reactions contained 10 µL of QX200 ddPCR EvaGreen Supermix (Bio-Rad Laboratories), 100 nM of each primer, 15 ng of genomic DNA, 4 units of *Hin*dIII restriction enzyme, and distilled water to a volume of 20 µL. PCR mixtures were separated into droplets by a Bio-Rad QX200 Droplet Generator. PCR amplification was performed in a Bio-Rad C1000 Thermal Cycler with the following PCR conditions: 95 °C for 7 min, 40 cycles of 30 s at 94 °C, 30 s at 58 °C and 60 s at 72 °C, followed by signal stabilization steps at 4 °C for 5 min and 90 °C for 5 min. After PCR amplification droplets were analyzed using a Bio-Rad QX200 Droplet Reader with Bio-Rad QuantaSoft software version 1.4. The ratio of positive to total droplets of the target gene was normalized to actin. Three biological replicates were used by independently extracting gDNA from different plants. Copy number variation in transgenic plants carrying *Mla1*, *Mla6*, and *Mla8* was determined by iDna Genetics (Norwich, UK) by quantitative real time PCR using the selectable marker gene *hyg* similar to the approach by Bartlett et al.[106].

**Reporting summary**. Further information on research design is available in the Nature Research Reporting Summary linked to this article.

## Data availability
The RNAseq data generated in this study have been deposited in the NCBI database under BioProject accessions PRJNA292371, PRJNA376252, PRJNA378334, and PRJNA378723. The NLR gene captures of barley accessions CI 16153 and Golden Promise generated in this study have been deposited in the NCBI database under BioProject accessions PRJNA523805 and PRJNA523807, respectively. The sequences of plasmids used for plant transformation are available in the source data file for Supplementary Fig. 3, and Figshare[107]. Genotypic, phenotypic, and raw data for figures and supplementary figures have been deposited on Figshare[107,108]. A material transfer agreement with The Sainsbury Laboratory is required to receive any materials. The use of the materials will be limited to non-commercial research uses only. Please contact M.J.M. (matthew.moscou@tsl.ac.uk) regarding the transgenic materials, and requests will be responded within 60 days. Source data are provided with this paper.

## Code availability
The QKcartographer, QKgenome, and QKphylogeny suite of Python scripts are maintained on GitHub [https://github.com/matthewmoscou/QKcartographer; https://github.com/matthewmoscou/QKgenome; https://github.com/matthewmoscou/QKphylogeny] and Figshare[107,109–111].

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

## Acknowledgements

The authors greatly appreciated valuable discussions with Peter van Esse, Jonathan Jones, Sophien Kamoun, Nick Talbot, and Cyril Zipfel. Photography was supported by Andrew Davis and Phil Robinson. Assistance in the greenhouse was provided by Sue Banfield and the John Innes Horticultural team. Seed was kindly provided by Andy Kleinhofs, Rients Niks, Ralph Panstruga, Mapi Valles, Wolfgang Spielmeyer, Nils Stein, Wendy Harwood, Wheat Genetics Resource Center (Kansas State University), Patrick Hayes, Kazuhiro Sato, Roger Wise, and the National Small Grains Collection (USDA-ARS). Funding for this research includes Biotechnology and Biological Sciences Research Council Doctoral Training Programme (grant no. BB/F017294/1 to J.B. and M.J.M.) and Institute Strategic Programme (grant no. BB/J004553/1 to B.B.H.W., BB/P012574/1 to M.J.M., and BBS/E/J/000PR9795 to M.J.M.), Human Frontier Science Program Long-term Fellowship (grant no. LT000218/2011-L to M.J.M.), John Innes Foundation (rotation PhD studentship to PE), 2Blades Foundation, and Gatsby Charitable Foundation.

## Author contributions

Design of the research: J.B., I.H.P., A.M.D., R.B., B.B.H.W., A.D., E.R.W., and M.J.M.; performing the research: J.B., I.H.P., A.M.D., M.G., P.G., J.T., M.S., J.N.F., P.E., A.H., J.C., B.S., A.D., and M.J.M.; data analysis: J.B., I.H.P., A.M.D., M.S., R.W., A.D., and M.J.M.; manuscript writing: J.B., I.H.P., A.M.D., and M.J.M.

## Competing interests

Patent application PCT/US2016/060101 filed by M.J.M. and A.M.D. that encompasses the map-based cloning and identification of the candidate gene for *Rps6*. The remaining authors declare no competing interests.
