## [Peer Review File · Nature Communications]

The barley immune receptor Mla recognizes multiple pathogens and contributes to host range dynamicsReviewers' Comments:

Reviewer #1:

Remarks to the Author:

Betgenhaeuser et al. report that a subset of barley accessions is susceptible to an isolate (08/21) of the wheat stripe rust pathogen, *Puccinia striiformis* forma *specialis* *tritici* (Pst). This is notable because there is evidence for host-specific adaptation of Pst to wheat and of *Puccinia striiformis* f sp *hordei* (Psh) to the sister host species, barley. Thus, formally barley can be considered a non-host for Pst. The authors show that non-host resistance in barley to Pst isolate 08/21 is quantitatively determined by three loci, designated Rps6, Rps7 and Rps8. This conclusion is based on the segregation of infection phenotypes of this Pst strain on progeny of a doubled-haploid mapping population derived from a cross between susceptible SusPtrit and resistant Golden Promise (GP) barley accessions. Rps7 was found to co-segregate with the barley Mla disease resistance locus, known to condition race-specific immunity to the barley powdery mildew pathogen *Blumeria graminis* f sp *hordei* (Bgh), indicating this locus might mediate dual resistance to unrelated powdery mildew and rust fungal pathogens. Using transgenic barley expressing Mla8 the authors then show that two independently generated transgenic lines are indeed immune to Pst isolate 08/21 and retain susceptibility to a Psh isolate. The authors conclude that widespread introduction of wild germplasm in plant breeding programs, including Mla resistance specificities to Bgh, poses a risk to maintenance of host species specificity to Pst (see title of manuscript), eroding non-host resistance to this pathogen.

Although this work reports in principle an important discovery, i.e. barley Mla8 functions as non-host resistance determinant to Pst, the manuscript suffers from inappropriate extrapolations and remains almost incomprehensible to a readership outside of the plant breeding hemisphere. A key inference of this study is not supported by data (see manuscript title). Despite my strong criticism of the present manuscript, I hope a radical surgical intervention as outlined below can help focus this work on its essentials, eliminate premature generalizations and drastically improve readability.

Major questions and points of concern

1. First results paragraph: "Cultivar and landrace barley accessions were highly resistant to wheat stripe rust, which contrasted sharply with the frequency of susceptibility observed in the wild accessions (Extended Data Fig. 1)." The generalization "wheat stripe resistance" is inappropriate because infection phenotypes of only one isolate (08/21) are presented in Supplementary Table 1. Any inference that cultivated or landrace barley is 'highly resistant' and 'contrasted sharply' with the frequency of susceptibility is inappropriate without information on the population structure of the tested wild barley accessions. No information is provided whether the wild barley lines tested are derived from the same or different populations. There are numerous fully resistant wild barley accessions and for this reason it is critical to know whether the susceptible accessions are derived from one or several populations. The general statement mentioned above is also inappropriate because infection phenotypes are presented only for one Pst isolate, 08/21. Whether cultivar and landrace barley accessions are more resistant to Pst compared to wild accessions necessitates infection tests with a panel of Pst strains that represent the genetic diversity of Pst at the pathogen population level (see also point #3 below). Without this information it remains unclear whether the infection pattern detected with strain 08/21 on the tested barley accessions is unique to this strain or is commonly seen in interactions with genetically polymorphic strains collected from Pst pathogen population(s). Lack of information on the genetic diversity of Pst and on the populations structure of the tested wild barley accessions (see below) does not permit the authors to make general conclusions on the genetic architecture of barley to the Basidiomycete pathogen. Hence, general conclusions cannot be drawn on the 'maintenance of host species specificity' of immune receptors to Pst infection (see manuscript title).

2. Fig. 1 I am unable to retrieve from Fig. 1 or Supplementary Table 2 the total number of DH progeny

that have been tested for resistance or susceptibility to Pst 08/21 and are derived from the cross SusPrit x GP. Were inoculation experiments with the Pst 08/21 strain at least once validated by a full factorial replicate to assess the robustness of infection phenotypes and infection categories shown in Fig. 1? Given that genome-wide information on molecular markers of this DH population is apparently available, I suggest the authors apply a more transparent and less biased visualization of the mapping results and display LOD scores along the five barley chromosomes. The visualization shown in Fig. 1 is in my view inappropriate for transparent data reporting. Is it correct that LOD scores have been calculated in Suppl. Table 2 for a select number of DH progeny?

3. Fig. 2 shows useful information on the prevalence of Rps6, Rps7, and Rps8 throughout barley accessions. However, it is a missed opportunity that progeny of the respective barley populations were tested only for reaction to Pst 08/21. Examination of the same barley lines with a number of Pst isolates collected from different geographic locations will reveal whether Rps6, Rps7, and Rps8 define unique QTL resistance loci to Pst 08/21 or represent major resistance QTLs to several or even the majority of Pst strains present in the pathogen population. This information is critical to make any generalized statement on the genetic architecture of barley resistance to Pst (see manuscript title and abstract).

4. Why were 5' regulatory sequences of Mla6 used to drive the expression of Mla8 in transgenic barley? This defies logic. One would have expected that Mla6 is driven by native Mla6 regulatory sequences in the transgenic plants. This necessitates a further control for data shown in Fig. 3: validation with two Bgh strains that differ in the presence or absence of AVRA8. This control will reveal whether the transgenic barley Mla8 plants retain the expected Mla8-specified strain-specific immunity to the powdery mildew pathogen Bgh. Information needs to be provided also on the length of the 5' and 3' regulatory sequences used to generate the p6:Mla8:t6 construct.

5. If I interpret the text in the results section and data in the Supplementary Tables correctly, Mla7 can also condition Rps7 resistance to Pst 08/21? If this is the case, why did the authors not generate transgenic barley lines expressing a p6:Mla7:t6 construct for functional validation? If this potentially exciting observation were true, does this indicate Pst 08/21 harbors one stripe rust effector recognized by both Mla7 and Mla8 or does this Pst strain harbor two effector genes of which one is recognized by Mla7 and the other by Mla8? I realize that the latter point is beyond the scope of the present work, but at the minimum this needs to be an integral part of the Discussion section.

6. Fig. 3. I am confused that the authors used a different scoring system for infection phenotypes in Fig. 3 compared to Fig. 1. This makes it impossible to cross-reference data shown both Figures. Why is quantitative information on pCOL (%) and pPUST (%) not integrated in Fig. 3?

7. The current manuscript is exceptionally compact for Nature Communications manuscripts. I presume this reflects a potential manuscript transfer from another Nature sister journal. In case the authors cannot clarify whether Rps6, Rps7, and Rps8 represent a unique disease resistance architecture to Pst strain 08/21 or have broader significance in non-host resistance to Pst at the pathogen population level, I recommend to reframe the manuscript, including manuscript title, by emphasizing dual Mla function in non-host resistance to Pst strain 08/21 and race-specific immunity to host-adapted Bgh. This will provide opportunities to discuss population genetic and mechanistic models how perhaps only a subset of barley Mla resistance specificities to Bgh have evolved with dual function in non-host resistance to Pst.

Minor point

1. Do the data shown in Extended Fig. 2 (droplet PCR) indicate true variation in Mla8 gene copies among the tested Mla8-containing barley accessions (~1.5 gene copies) or is this variation merely intrinsic 'noise' of the assay? At least the tested transgenic barley lines T1-6 and T1-7 appear to

harbor a single (hemizygous) copy of Mla8.

Reviewer #2:

Remarks to the Author:

This is a very interesting paper. It is of great interest that the same R-gene would confer resistance to barley powdery mildew and wheat stripe rust, and maybe even to more pathogens, like to causal agents of spot blotch and rice blast. This is new information, and helps us understanding some aspects of the genetics of nonhost resistance in plants. Nonhost resistance is a phenomenon occurring in all plant and animal species and its genetic basis is still poorly understood. In this submission, the data seem to have been collected and are presented in a competent way, but see my comments to convince even more. Also I find the practical conclusions/recommendations rather overstated, as explained below.

My most important point is that the paper lacks data on barley powdery mildew infection results on the transgenic lineages depicted in Figure 3 and Extended data figure 3. It would have been much more convincing if the positive transformants would have been shown not only be resistant to wheat stripe rust but also to powdery mildew. If the results would have been negative (so, the Mla8 would not be effective to powdery mildew) I would like to have an explanation.

The authors make a big issue of the danger of causing host jumps, by introduction of genes from wild accessions. The Pc2- Crown rust case is a good example: in that case resistance to one pathogen species was associated with susceptibility to the other. I find this in the context of the present paper a rather "empty" warning, since 1) it cannot be predicted which transferred gene may confer susceptibility to a novel pathogen. 2) In contrast to the Pc2 – crown rust example, the gene Rps7/Mla8 confers resistance to two pathogens (Pst and Bgt) at the same time. So this case suggests that transfer of a powdery mildew resistance gene might at the same time enhance the resistance to stripe rust? So: should breeders refrain categorically from using exotic donors for resistance? I would not think so.

The authors claim that that wild accessions may be susceptible to unadapted pathogens. Well, their data suggests indeed a somewhat higher proportion of susceptible items in the wild material (still less than 25% in terms of sporulation) than in cultivated accessions, but 1) only sporulation counts, since chlorosis will not result in an epidemic, and 2) more importantly, do the authors expect/know whether adult plants are equally susceptible as the seedlings? Are their most susceptible genotypes also susceptible as the plants develop further? There may be redundancy of genes for nonhost resistance, especially at the adult plant stage? Susceptibility at seedling stage is epidemiologically less relevant than susceptibility at advanced development stages. There is much evidence that adult plants tend to have much stronger nonhost resistance than seedlings.

It would be helpful to provide the statistics on prevalence of the susceptibility to wheat stripe rust compared to landraces and cultivars. In Extended data figure I count 25 wild accessions and one in the 2-row landraces (blue name there) of which only six allow sporulation (about 25%). Therefore I consider "Cultivar and landrace barley accessions were highly resistant to wheat stripe rust, which contrasted sharply with the frequency of susceptibility observed in the wild accessions" quite an overstatement. Still the large majority of the wild barleys do not allow sporulation.

It is suggested that Rps6, 7 and 8 together greatly contribute to the nonhost status of barley to the wheat stripe rust. Still I cannot follow the text on the presence of these genes: "Rps7 was detected only in the wild accession WBDC172." Maybe this should read as: "Among the wild barley accessions, WBDC172 was the only one in which Rps7 was detected"? A few lines above, the authors state that Rps7 was found in "56% (14/25) of the accessions analysed." , so in more items than only WBDC172. In figure 2 I count only 13 accessions with some orange coloured pie out of 24(!) investigated

accessions: so 13/24? What is in that figure the meaning of "Morex leaf" versus "Morex"?

About Rps6: "However, it was detected with large effect sizes in all but one wild accession." This sentence is multi-interpretable. Always with large effect size, but a small effect size in only one wild accession? (but Haruna Nijo, which is one of TWO where I see a small effect, is no wild accession?). Or: With large effect size in all wild accessions, except for CIho 4196, where the effect was small? Or: always large effect size, but it was absent in one wild accession? In figure 2 it is not easy to see which accession is wild (I should compare with Supp Table 1, I suppose).

"Rps8 functioned independently from Rps7 and Rps6 in three accessions with effect sizes ranging from 22 to 31% PVE for colonization and 12 to 60% for pustule formation resistance." I would expect there that "in three accessions the Rps8 gene occurred in absence of Rps6 and Rps7." "functioned independently" suggests that the two other genes did occur, but had no effect?

"RGH3 was found to be present using comprehensive sequence captures targeting the entire Mla locus of accessions Golden Promise and CI 16153, although RGH3 was not expressed in accessions harbouring Mla7 or Mla8." So, Golden Promise contains Mla8, and has at least one copy of RGH3, although RGH3 was not expressed in accessions carrying Mla7 or Mla8? Why not: "... but not expressed in those (two) accessions"? Can you generalize for all Mla7 and Mla8 accessions?

"Mla is a target of breeding, often through the introduction of exotic alleles from wild barley, many of which do not confer resistance to wheat stripe rust [37]." The reference 37 does not report anything about resistance/susceptibility of the wild barley accessions to wheat stripe rust. That paper even does not report any result of rust and also did not particularly focus on Mla. How do the authors know that "many of which (= Mla alleles? Ml genes in general?) do NOT confer resistance to wheat stripe rust?"

Some typo's and minor issues:

...that exhibit host ...= that exhibits host ...

"no pustules observed" ..."covered in pustules". Powdery mildew is generally not considered to form pustules (like rust), but (mycelial) colonies.

"... using the flanking markers K_963924 and K_206D11." I did not see a reference or these markers being mentioned in, for example, figure 3a.

"...DNA from positives colonies were ..." should read "positive"?

Reviewer #3:

Remarks to the Author:

This manuscript describes that barley non-host resistance to wheat stripe rust primarily is mediated by three resistance genes (Rps6, Rps7 and Rps8), and that Rps7 is identical to the powdery mildew resistance gene, Mla. This result was obtained by an initial mapping in a Golden Promise x SusPtrit mapping population followed by studies of several populations in which one or more of the three genes segregate. Fine-mapping suggested Rps7 to be Mla, which was confirmed by overexpression. A number of the known Mla alleles that confer different race-specific resistances to powdery mildew, all conferred resistance to wheat stripe rust (see below).

A large amount of work is presented and the results are highly important. The work indicates that the non-host resistance of barley to wheat stripe rust is really fragile as many barley wild accessions are susceptible, while some cultivated genotypes are protected by only one resistance gene. It also shows that Rps7, which is the most essential of the three genes, is having the risk of becoming non-

functional as breeders are introducing new alleles of this gene in order to protect against powdery mildew. Another implication of this risk of having wheat stripe rust on barley is that this may worsen the disease pressure on wheat as the global inoculum pressure will increase.

Having said this, a number of issues should be addressed as some of the presentation is challenging to follow.

Fig. 2. It really took a lot of effort to try to understand Fig. 2. Please explain "Tree scale = 1" relative to the number of SNPs. What is the link? In fact, the phylogenetic analysis is not used and the question is how important it is to show it in Fig. 2? This figure would be much easier to grasp with histograms rather than pie charts, and the phylogenetic analysis could be provided as supplemental data. In the text, there are some numbers that does not appear to match the figure. It says "Rps7 was observed in 56% (14/25) of the accessions". Apparently, this should be 13/24. Please correct or explain. It also says "In five instances, Rps7 was detected independent of Rps6 and Rps8". Shouldn't it be "In four instances, Rps7 was detected independentLY of Rps6 and Rps8" ?

Maybe rewrite the text "Rps7 was detected only in the wild barley accession WBDC172, suggesting" to "WBDC172 was the only in wild barley accession in which Rps7 was detected, suggesting".

The present use of the term "Mla locus" to describe a larger chromosome interval with several RGH's clearly has historical reasons. However, it appears not to be relevant here. Why not restrict the study to say something like "Eight and three recombinations (I take this is what the numbers in Fig. 3a indicate) were found between Rps7 and the closest genetic markers on either side, which is in agreement with the position of the Mla gene. This candidate prediction was tested in transgenic plants". It is not clear what "suppressed recombination" refers to. Is it simply that the physical distance is small, leaving little chance for recombination, or does it reflect actual interference with recombination. Maybe leave out this phrasing to avoid confusion. The purposes of mentioning RGH2 and RGH3 in the test, and the drawings of the RGH1 and RGH2 contigs in Fig. 3B are not clear.

Extended Data Fig. 3 is hard to understand. Does it show box plots? Why does the transgenic lines show so variable results? Due to segregation?

It is highly interesting that Rps7 and Mla is one and the same gene. However, it is not entirely clear which of the well-known alleles of this gene provides resistance to wheat stripe rust, although it appears that many of them do. Is it so that all known alleles that encodes a full-length NLR provide wheat stripe rust resistance? It seems that the data are available. Please be more explicit and perhaps provide a supplemental data file to document this.

It appears that the wheat stripe rust resistance of some barley genotypes is based on a single R-gene. Please address this briefly, and comment on the fact that this resistance remains functional. One could have expected it to be broken long time ago.

General response to the reviewers: Thank you for your time and effort in critically commenting on our manuscript. We have substantially revised the manuscript and incorporated the majority of the reviewers' suggestions. A summary of new experiments performed include:

1. Phenotyping of two diverse populations of 196 elite and 313 wild accessions with *P. striiformis* f. sp. *tritici* isolate 08/21.
2. Quantification of copy number variation of *Mla7* and *Mla8* in diverse barley accessions.
3. The generation of independent transformants for *Mla1*, *Mla6*, and *Mla7* expressed under the *Mla6* promoter/terminator.
4. Phenotypic assessment of *Mla1*, *Mla6*, *Mla7*, and *Mla8* transformants with 14 isolates of *B. graminis* f. sp. *hordei* and *P. striiformis* f. sp. *tritici* isolate 16/035.

Reviewer #1 (Remarks to the Author):

Bettgenhaeuser et al. report that a subset of barley accessions is susceptible to an isolate (08/21) of the wheat stripe rust pathogen, *Puccinia striiformis* forma specialis *tritici* (*Pst*). This is notable because there is evidence for host-specific adaptation of *Pst* to wheat and of *Puccinia striiformis* f sp *hordei* (*Psh*) to the sister host species, barley. Thus, formally barley can be considered a non-host for *Pst*. The authors show that non-host resistance in barley to *Pst* isolate 08/21 is quantitatively determined by three loci, designated *Rps6*, *Rps7* and *Rps8*. This conclusion is based on the segregation of infection phenotypes of this *Pst* strain on progeny of a doubled-haploid mapping population derived from a cross between susceptible SusPtrit and resistant Golden Promise (GP) barley accessions. *Rps7* was found to co-segregate with the barley *Mla* disease resistance locus, known to condition race-specific immunity to the barley powdery mildew pathogen *Blumeria graminis* f sp *hordei*.

(*Bgh*), indicating this locus might mediate dual resistance to unrelated powdery mildew and rust fungal pathogens. Using transgenic barley expressing *Mla8* the authors then show that two independently generated transgenic lines are indeed immune to *Pst* isolate 08/21 and retain susceptibility to a *Psh* isolate. The authors conclude that widespread introduction of wild germplasm in plant breeding programs, including *Mla* resistance specificities to *Bgh*, poses a risk to maintenance of host species specificity to *Pst* (see title of manuscript), eroding non-host resistance to this pathogen.

Although this work reports in principle an important discovery, i.e. barley *Mla8* functions as non-host resistance determinant to *Pst*, the manuscript suffers from inappropriate extrapolations and remains almost incomprehensible to a readership outside of the plant breeding hemisphere. A key inference of this study is not supported by data (see manuscript title). Despite my strong criticism of the present manuscript, I hope a radical surgical intervention as outlined below can help focus this work on its essentials, eliminate premature generalizations and drastically improve readability.

Response: We appreciate the criticism raised by the reviewer and have refocused the manuscript, including its title, around multiple pathogen recognition by *Mla* and its contribution to host range dynamics.

Major questions and points of concern

1. First results paragraph: "Cultivar and landrace barley accessions were highly resistant to wheat stripe rust, which contrasted sharply with the frequency of susceptibility observed in the wild accessions (Extended Data Fig. 1)." The generalization "wheat stripe resistance" is

inappropriate because infection phenotypes of only one isolate (08/21) are presented in Supplementary Table 1. Any inference that cultivated or landrace barley is ‘highly resistant’ and ‘contrasted sharply’ with the frequency of susceptibility is inappropriate without information on the population structure of the tested wild barley accessions. No information is provided whether the wild barley lines tested are derived from the same or different populations. There are numerous fully resistant wild barley accessions and for this reason it is critical to know whether the susceptible accessions are derived from one or several populations. The general statement mentioned above is also inappropriate because infection phenotypes are presented only for one *Pst* isolate, 08/21. Whether cultivar and landrace barley accessions are more resistant to *Pst* compared to wild accessions necessitates infection tests with a panel of *Pst* strains that represent the genetic diversity of *Pst* at the pathogen population level (see also point #3 below). Without this information it remains unclear whether the infection pattern detected with strain 08/21 on the tested barley accessions is unique to this strain or is commonly seen in interactions with genetically polymorphic strains collected from *Pst* pathogen population(s). Lack of information on the genetic diversity of *Pst* and on the populations structure of the tested wild barley accessions (see below) does not permit the authors to make general conclusions on the genetic architecture of barley to the Basidiomycete pathogen. Hence, general conclusions cannot be drawn on the ‘maintenance of host species specificity’ of immune receptors to *Pst* infection (see manuscript title).

Response: In our work, we had a critical decision to invest our efforts either into the evaluation of the genetic diversity of barley or alternatively, the genetic diversity of *Pst*. In part, our ability to work with *Pst* from diverse regions in the world is associated with limited containment facilities to house foreign isolates. We agree with the reviewer that we are limited in our broader claim and have modified this in the text accordingly. We add that this work, in particular the generation of stable transgenic accessions, provides genetic material for the future assessment with international collections.

The majority of wild barley accessions (N=20) used in this study are a subset of accessions derived from the Wild Barley Diversity Collection (WBDC) curated by Prof. Brian Steffenson (University of Minnesota) and Dr. Jan Valkoun (ICARDA). All accessions have geospatial information and have been genotyped using GBS, which was used to determine population structure. Sallem *et al.* (2017) reported a total of eight populations based on 314 WBDC accessions. These selected accessions belong to seven of eight populations (SP1 (N=4), SP2 (N=3), SP4 (N=1), SP5 (N=3), SP6 (N=1), SP7 (N=7), and SPN (N=6)). It should be highlighted that no previous study has incorporated this degree of genetic diversity in barley and its interaction with *P. striiformis* f. sp. *tritici*. To provide greater support for our claim, we extended our work by including the entire population (313 accessions), as well as a collection of 2-row barley accessions (AGEOUB panel; N=196). A description of these results has been added to the main text and four histograms in Extended Data Fig. 1.

In this manuscript, we use four isolates of *Puccinia striiformis*. These include *P. striiformis* f. sp. *tritici* (*Pst*) isolates 08/21, 15/151, and 16/035 and *P. striiformis* f. sp. *hordei* (*Psh*) isolate B01/2. The relationship of *Pst* isolate 08/21 and *Psh* B01/2 is detailed in Bettgenhaeuser *et al.* PLoS Genetics (2018). *Pst* 08/21 belongs to the ‘Old UK/France’ lineage (Hubbard *et al.* (2015) Genome Research; Bueno-Sancho *et al.* (2017) Genome Biol. Evol.), whereas *Pst* 15/151 and 16/035 belong to the ‘Europe Warrior’ lineage (UKCVPS Annual Report 2015; 2016; Diane Saunders, personal communication). In our experiments with barley, we did not observe phenotypic variation between these isolates for their recognition by *Rps6*, *Rps7*, or *Rps8*. We agree that future work is necessary to see if this holds based on worldwide diversity.

2. Fig. 1 I am unable to retrieve from Fig. 1 or Supplementary Table 2 the total number of DH progeny that have been tested for resistance or susceptibility to Pst 08/21 and are derived from the cross SusPrit x GP. Were inoculation experiments with the Pst 08/21 strain at least once validated by a full factorial replicate to assess the robustness of infection phenotypes and infection categories shown in Fig. 1? Given that genome-wide information on molecular markers of this DH population is apparently available, I suggest the authors apply a more transparent and less biased visualization of the mapping results and display LOD scores along the five barley chromosomes. The visualization shown in Fig. 1 is in my view inappropriate for transparent data reporting. Is it correct that LOD scores have been calculated in Suppl. Table 2 for a select number of DH progeny?

Response: In the initial submission, we provided all raw phenotypic and genotypic data in a Figshare repository in the 'Data availability' section. The link is <https://www.doi.org/10.6084/m9.figshare.7763018>. This supplemental data includes genetic maps, phenotypic distributions, pairwise plots of macroscopic and microscopic phenotypes, and QTL analyses for each replicate data set. Size of all populations used in this study are described in Supplemental Table 5. The SxGP DH population was inoculated and phenotyped in two independent replicates, with both experiments identifying the same three QTL (*Rps6*, *Rps7*, and *Rps8*), as well as HxO DH and FxC RIL population. In Supplemental Table 2, the QTL results are for the average of the two replicates for each population and are based on the use of the entire population. This was not clear; therefore we have added text to reflect this. Individual QTL analyses are provided in the Supplemental Data. We have added the composite interval mapping results to Fig. 1, panel A. We have retained the phenotype x genotype plot as Panel B, as it shows the simple genetic architecture of three major effect QTLs. We believe this is appropriate, as no other major effect QTL were detected and the majority of the cumulative phenotypic variation is explained by these three loci (PVE > 61% for chlorosis and PVE > 30% for infection). All information reported in Supplemental Table 2 is based on the entire population and are the formatted output from QTL Cartographer.

3. Fig. 2 shows useful information on the prevalence of *Rps6*, *Rps7*, and *Rps8* throughout barley accessions. However, it is a missed opportunity that progeny of the respective barley populations were tested only for reaction to Pst 08/21. Examination of the same barley lines with a number of Pst isolates collected from different geographic locations will reveal whether *Rps6*, *Rps7*, and *Rps8* define unique QTL resistance loci to Pst 08/21 or represent major resistance QTLs to several or even the majority of Pst strains present in the pathogen population. This information is critical to make any generalized statement on the genetic architecture of barley resistance to Pst (see manuscript title and abstract).

Response: We share the enthusiasm of the reviewer in determining whether the genetic architecture identified in this work can broadly explain host range at a global scale. We believe this is beyond the scope of the current work and tempered the text to reflect this. In future work, the identification of *Rps6*, *Rps7* (this work), and *Rps8* and generation of stable transgenic lines in a consistent genetic background will facilitate worldwide phenotyping with collaborators.

4. Why were 5' regulatory sequences of *Mla6* used to drive the expression of *Mla8* in transgenic barley? This defies logic. One would have expected that *Mla6* is driven by native *Mla6* regulatory sequences in the transgenic plants. This necessitates a further control for data shown in Fig. 3: validation with two *Bgh* strains that differ in the presence or absence of AVRA8. This control will reveal whether the transgenic barley *Mla8* plants retain the expected

Mla8-specified strain-specific immunity to the powdery mildew pathogen Bgh. Information needs to be provided also on the length of the 5' and 3' regulatory sequences used to generate the p6:Mla8:t6 construct.

Response: We have generated stable transgenic lines expressing *Mla1*, *Mla6*, and *Mla7* using the *Mla6* promoter. These have been tested with a collection of 13 different isolates that vary in presence or absence in the corresponding effector (see Fig. 4). We show that the *Mla6* promoter is sufficient to express *Mla1*, *Mla6*, and *Mla8*, as well as retain their specificity to barley powdery mildew and wheat stripe rust. The *Mla6* promoter was insufficient to drive *Mla7* expression as a single copy, although multicopy lines exhibited resistance and specificity to barley powdery mildew and wheat stripe rust. Interestingly, *Mla7* natively exists in multiple copies (Extended Data Fig. 3), suggesting that the *Mla7* promoter may be insufficient to drive expression as a single copy. We have added Supplemental Fig. 1 showing the promoter sequence used from *Mla6* and the inserts for constructs carrying *Mla1*, *Mla6*, *Mla7*, and *Mla8*.

5. If I interpret the text in the results section and data in the Supplementary Tables correctly, Mla7 can also condition Rps7 resistance to Pst 08/21? If this is the case, why did the authors not generate transgenic barley lines expressing a p6:Mla7:t6 construct for functional validation? If this potentially exciting observation were true, does this indicate Pst 08/21 harbors one stripe rust effector recognized by both Mla7 and Mla8 or does this Pst strain harbor two effector genes of which one is recognized by Mla7 and the other by Mla8? I realize that the latter point is beyond the scope of the present work, but at the minimum this needs to be an integral part of the Discussion section.

Response: As mentioned above, we have generated this transgenic line. The group of Paul Schulze-Lefert has identified AVRa7, and while AVRa8 is not identified, the current evidence suggests it is a completely different gene (similar to observations with other AVRa genes). We agree with the reviewer that our current working hypothesis is that *Mla7* and *Mla8* likely recognize different effectors in wheat stripe rust. We have incorporated text to discuss this hypothesis and the mechanism of recognition by Mla in the Discussion section.

6. Fig. 3. I am confused that the authors used a different scoring system for infection phenotypes in Fig. 3 compared to Fig. 1. This makes it impossible to cross-reference data shown both Figures. Why is quantitative information on pCOL (%) and pPUST (%) not integrated in Fig. 3?

Response: In our earlier work, we incorporated pCOL and pPUST to ensure that chlorosis and infection were consistently correlated. As the work progressed, it became clear that these traits are always correlated and there was limited value in collecting this information.

7. The current manuscript is exceptionally compact for Nature Communications manuscripts. I presume this reflects a potential manuscript transfer from another Nature sister journal. In case the authors cannot clarify whether Rps6, Rps7, and Rps8 represent a unique disease resistance architecture to Pst strain 08/21 or have broader significance in non-host resistance to Pst at the pathogen population level, I recommend to reframe the manuscript, including manuscript title, by emphasizing dual Mla function in non-host resistance to Pst strain 08/21 and race-specific immunity to host-adapted Bgh. This will provide opportunities to discuss population genetic and mechanistic models how perhaps only a subset of barley Mla resistance specificities to Bgh have evolved with dual function in non-host resistance to Pst.

Response: Based on these comments, and those of the other reviewers, we altered the emphasis of this manuscript and expanded elements of the manuscript accordingly.

Minor point

1. Do the data shown in Extended Fig. 2 (droplet PCR) indicate true variation in *Mla8* gene copies among the tested *Mla8*-containing barley accessions (~1.5 gene copies) or is this variation merely intrinsic ‘noise’ of the assay? At least the tested transgenic barley lines T1-6 and T1-7 appear to harbor a single (hemizygous) copy of *Mla8*.

Response: This figure is now Extended Data Fig. 3. The reviewers’ comments prompted an experiment where we independently extracted gDNA and performed the droplet PCR assay. The assay itself appears to create the reduced estimate of 1.5 versus the expected 2.0 copies. Progeny of transgenic barley lines T1-6 and T1-7 were tested with a qPCR assay from a service provider (iDna Genetics) and confirmed to segregate for a single T-DNA. An additional assay was developed for *Mla7* and shows higher copy number variation in this gene.

Reviewer #2 (Remarks to the Author):

This is a very interesting paper. It is of great interest that the same R-gene would confer resistance to barley powdery mildew and wheat stripe rust, and maybe even to more pathogens, like to causal agents of spot blotch and rice blast. This is new information, and helps us understanding some aspects of the genetics of nonhost resistance in plants. Nonhost resistance is a phenomenon occurring in all plant and animal species and its genetic basis is still poorly understood. In this submission, the data seem to have been collected and are presented in a competent way, but see my comments to convince even more. Also I find the practical conclusions/recommendations rather overstated, as explained below.

Response: Based on your comments and those from the other reviewers, we have altered the manuscript to refocus on the dual capacity of *Mla* to recognize barley powdery mildew and wheat stripe rust.

My most important point is that the paper lacks data on barley powdery mildew infection results on the transgenic lineages depicted in Figure 3 and Extended data figure 3. It would have been much more convincing if the positive transformants would have been shown not only be resistant to wheat stripe rust but also to powdery mildew. If the results would have been negative (so, the *Mla8* would not be effective to powdery mildew) I would like to have an explanation.

Response: Please see the response to Reviewer 1, Point 4. We believe we have addressed these concerns through additional experimentation (including transgenic lines carrying *Mla1*, *Mla6*, and *Mla7*).

The authors make a big issue of the danger of causing host jumps, by introduction of genes from wild accessions. The Pc2- Crown rust case is a good example: in that case resistance to one pathogen species was associated with susceptibility to the other. I find this in the context of the present paper a rather “empty” warning, since 1) it cannot be predicted which transferred gene may confer susceptibility to a novel pathogen. 2) In contrast to the Pc2 – crown rust example, the gene *Rps7/Mla8* confers resistance to two pathogens (*Pst* and *Bgt*) at the same time. So this case suggests that transfer of a powdery mildew resistance gene might at the same

time enhance the resistance to stripe rust? So: should breeders refrain categorically from using exotic donors for resistance? I would not think so.

Response: We appreciate the comments of the reviewer and have emphasized the dual recognition of *Mla* as the emphasis of the manuscript. *Pc2* is another example of dual recognition, albeit with an outcome of resistance or susceptibility. The *Mla* locus has also been associated with susceptibility to spot blotch. We do not suggest that breeders refrain from using exotic donors, but we highlight a risk. This work shows how existing genes that may not be perceived as contributing to a function, in fact do have a function. The classical example is *Mla8*, which was believed by plant pathologists in Europe to provide no value for barley due all known European isolates being virulent on this gene, but we now know this gene contributes to maintaining host species specificity to non-adapted stripe rust.

The authors claim that that wild accessions may be susceptible to unadapted pathogens. Well, their data suggests indeed a somewhat higher proportion of susceptible items in the wild material (still less than 25% in terms of sporulation) than in cultivated accessions, but 1) only sporulation counts, since chlorosis will not result in an epidemic, and 2) more importantly, do the authors expect/know whether adult plants are equally susceptible as the seedlings? Are their most susceptible genotypes also susceptible as the plants develop further? There may be redundancy of genes for nonhost resistance, especially at the adult plant stage? Susceptibility at seedling stage is epidemiologically less relevant than susceptibility at advanced development stages. There is much evidence that adult plants tend to have much stronger nonhost resistance than seedlings.

Response: We have addressed this comment by refocusing the manuscript. To our knowledge, only one study has been performed (Niks et al. (2013) European Journal of Plant Pathology) at comparing seedling and adult stage phenotypes. While beyond the scope of this manuscript, we agree that it will be an important result to determine the role of barley in the epidemiology of wheat stripe rust.

It would be helpful to provide the statistics on prevalence of the susceptibility to wheat stripe rust compared to landraces and cultivars. In Extended data figure I count 25 wild accessions and one in the 2-row landraces (blue name there) of which only six allow sporulation (about 25%). Therefore I consider “Cultivar and landrace barley accessions were highly resistant to wheat stripe rust, which contrasted sharply with the frequency of susceptibility observed in the wild accessions” quite an overstatement. Still the large majority of the wild barleys do not allow sporulation.

Response: We appreciate that the phrasing is inaccurate and have modified the text accordingly. Our desire was to contrast the extremely high levels of resistance observed in elite germplasm with the more frequent, albeit still low, levels of susceptibility in wild germplasm. We have expanded our analysis to include 313 wild barley accessions and found that approximately 32% (101 accessions) exhibited some degree of pustule formation. To our knowledge, this phenomenon has not been observed in the interaction of barley and wheat stripe rust.

It is suggested that Rps6, 7 and 8 together greatly contribute to the nonhost status of barley to the wheat stripe rust. Still I cannot follow the text on the presence of these genes: “Rps7 was detected only in the wild accession WBDC172.” Maybe this should read as: “Among the wild barley accessions, WBDC172 was the only one in which Rps7 was detected”? A few lines

above, the authors state that Rps7 was found in “56% (14/25) of the accessions analysed.” , so in more items than only WBDC172. In figure 2 I count only 13 accessions with some orange coloured pie out of 24(!) investigated accessions: so 13/24? What is in that figure the meaning of “Morex leaf” versus “Morex”?

Response: We have corrected this sentence referring to WBDC172. In Fig. 2, the accession HOR2926 was not included, as we do not have RNAseq data available. Supplementary Table 4 was used to create this summary information. In Fig. 2, Morex leaf is a control, where RNAseq from Morex leaf tissue was used for SNP calling in the generation of the phylogenetic tree. Text was added to the figure legend to make this clearer.

About Rps6: “However, it was detected with large effect sizes in all but one wild accession.” This sentence is multi-interpretable. Always with large effect size, but a small effect size in only one wild accession? (but Haruna Nijo, which is one of TWO where I see a small effect, is no wild accession?). Or: With large effect size in all wild accessions, except for CIho 4196, where the effect was small? Or: always large effect size, but it was absent in one wild accession? In figure 2 it is not easy to see which accession is wild (I should compare with Supp Table 1, I suppose).

Response: There was a mistake in this sentence, as two accessions have small effect sizes for *Rps6* (CIho 4196 and Haruna Nijo). We have changed this sentence to “When detected, *Rps6* had large effect sizes in the majority of accessions (9/11).” All wild barley accessions used for mapping are found on the right of the figure and have the prefix WBDC (as they are derived from the WBDC panel).

“Rps8 functioned independently from Rps7 and Rps6 in three accessions with effect sizes ranging from 22 to 31% PVE for colonization and 12 to 60% for pustule formation resistance.” I would expect there that “in three accessions the Rps8 gene occurred in absence of Rps6 and Rps7.” “functioned independently” suggests that the two other genes did occur, but had no effect?

Response: We changed ‘functioned independently’ to ‘observed in isolation’.

“RGH3 was found to be present using comprehensive sequence captures targeting the entire *Mla* locus of accessions Golden Promise and CI 16153, although RGH3 was not expressed in accessions harbouring *Mla7* or *Mla8*.” So, Golden Promise contains *Mla8*, and has at least one copy of RGH3, although RGH3 was not expressed in accessions carrying *Mla7* or *Mla8*? Why not: “ ... but not expressed in those (two) accessions”? Can you generalize for all *Mla7* and *Mla8* accessions?

Response: We have corrected the sentence in the following manner “Evaluation of RNAseq data found that *RGH3* was not expressed in the accessions with available data harbouring *Mla7* or *Mla8*”. We agree that this assumption may not be valid.

“*Mla* is a target of breeding, often through the introduction of exotic alleles from wild barley, many of which do not confer resistance to wheat stripe rust [37].” The reference 37 does not report anything about resistance/susceptibility of the wild barley accessions to wheat stripe rust. That paper even does not report any result of rust and also did not particularly focus on *Mla*. How do the authors know that “many of which (= *Mla* alleles? *Ml* genes in general?) do NOT confer resistance to wheat stripe rust?”

Response: This reference was a mistake between versions of the manuscript. The reference should be for Ames et al. (2015), who sought sources of resistance to barley powdery mildew in the wild barley WBDC panel. We have also added Jørgensen (1994) that specifically describes *Mla* alleles that were identified from wild barley that were introduced into elite germplasm. We have also rephrased the sentence.

Some typo's and minor issues:

...that exhibit host ...= that exhibits host ...

“no pustules observed” ...”covered in pustules”. Powdery mildew is generally not considered to form pustules (like rust), but (mycelial) colonies.

“... using the flanking markers K_963924 and K_206D11.” I did not see a reference or these markers being mentioned in, for example, figure 3a.

“...DNA from positives colonies were ...” should read “positive”?

Response: We have corrected the corresponding text. We have added the Foster x CIho 4196 RIL population genetic map to Fig. 3a to show the locations of K_963924 and K_206D11, whereas Fig. 3b now shows the recombinants immediately encompassing the *Mla/Rps7* locus. Text has also been integrated into the main text.

Reviewer #3 (Remarks to the Author):

This manuscript describes that barley non-host resistance to wheat stripe rust primarily is mediated by three resistance genes (*Rps6*, *Rps7* and *Rps8*), and that *Rps7* is identical to the powdery mildew resistance gene, *Mla*. This result was obtained by an initial mapping in a Golden Promise x SusPtrit mapping population followed by studies of several populations in which one or more of the three genes segregate. Fine-mapping suggested *Rps7* to be *Mla*, which was confirmed by overexpression. A number of the known *Mla* alleles that confer different race-specific resistances to powdery mildew, all conferred resistance to wheat stripe rust (see below).

A large amount of work is presented and the results are highly important. The work indicates that the non-host resistance of barley to wheat stripe rust is really fragile as many barley wild accessions are susceptible, while some cultivated genotypes are protected by only one resistance gene. It also shows that *Rps7*, which is the most essential of the three genes, is having the risk of becoming non-functional as breeders are introducing new alleles of this gene in order to protect against powdery mildew. Another implication of this risk of having wheat stripe rust on barley is that this may worsen the disease pressure on wheat as the global inoculum pressure will increase.

Having said this, a number of issues should be addressed as some of the presentation is challenging to follow.

Fig. 2. It really took a lot of effort to try to understand Fig. 2. Please explain “Tree scale = 1” relative to the number of SNPs. What is the link? In fact, the phylogenetic analysis is not used and the question is how important it is to show it in Fig. 2? This figure would be much easier

to grasp with histograms rather than pie charts, and the phylogenetic analysis could be provided as supplemental data. In the text, there are some numbers that does not appear to match the figure. It says “Rps7 was observed in 56% (14/25) of the accessions”. Apparently, this should be 13/24. Please correct or explain. It also says “In five instances, Rps7 was detected independent of Rps6 and Rps8”. Shouldn’t it be “In four instances, Rps7 was detected independently of Rps6 and Rps8” ?

Response: The phylogenetic tree in Fig. 2 is used to show the genetic relationship between the accessions evaluated for their resistance to wheat stripe rust. RNAseq was used as an unbiased approach for genotyping. Our goal is to show evidence that all three genes are detected throughout barley, including domesticated (elite and landrace) and wild barley. We appreciate that the figure is complex, but this is in part because it captures a substantial amount of information. The discrepancy between the text and figure for numbers was that the barley accession HOR 2926 did not have RNAseq data available, therefore it is excluded in the figure.

Maybe rewrite the text “Rps7 was detected only in the wild barley accession WBDC172, suggesting” to “WBDC172 was the only in wild barley accession in which Rps7 was detected, suggesting”.

Response: This sentence has been changed according to the reviewer’s suggestion.

The present use of the term “Mla locus” to describe a larger chromosome interval with several RGH’s clearly has historical reasons. However, it appears not to be relevant here. Why not restrict the study to say something like “Eight and three recombinations (I take this is what the numbers in Fig. 3a indicate) were found between Rps7 and the closest genetic markers on either side, which is in agreement with the position of the Mla gene. This candidate prediction was tested in transgenic plants”. It is not clear what “suppressed recombination” refers to. Is it simply that the physical distance is small, leaving little chance for recombination, or does it reflect actual interference with recombination. Maybe leave out this phrasing to avoid confusion. The purposes of mentioning RGH2 and RGH3 in the test, and the drawings of the RGH1 and RGH2 contigs in Fig. 3B are not clear.

Response: We have retained the use of the term ‘*Mla* locus’, as it reflects the suppressed recombination that is observed at the locus. In this instance, suppressed recombination refers to the observation of recombination events at the boundary of the locus and no recombination events within the locus. While not shown in the figure, there are >20 markers that cosegregate with *Mla* and *Rps7* in both populations. This is important, as it demonstrates our limited ability to use map-based cloning to further resolve the region and our reliance on other approaches (e.g. expression studies, transgenics) to further investigate the locus. We have updated Fig. 3 based on the recently released genomic sequence for barley cv. Golden Promise (*Mla8*). *RGH2* and *RGH3* are mentioned, as we had to establish whether these genes are present in the *Mla7* and *Mla8* haplotypes, and if so, if they are the causal gene to wheat stripe rust (rather than *Mla*).

Extended Data Fig. 3 is hard to understand. Does it show box plots? Why does the transgenic lines show so variable results? Due to segregation?

Response: We have updated this figure using color coding to differentiate between T-DNA positive and T-DNA negative lines. Transgenic lines showing a variable response were segregating for the T-DNA. We hypothesize that lines carrying the T-DNA but that show a

weak resistance response are lines hemizygous for *Mla8*. From the same transgenic family, advanced homozygous single gene inserts for *p6:Mla8:t6* provide full resistance to wheat stripe rust and barley powdery mildew (carrying *AVRa8*).

It is highly interesting that *Rps7* and *Mla* is one and the same gene. However, it is not entirely clear which of the well-known alleles of this gene provides resistance to wheat stripe rust, although it appears that many of them do. Is it so that all known alleles that encodes a full-length NLR provide wheat stripe rust resistance? It seems that the data are available. Please be more explicit and perhaps provide a supplemental data file to document this.

Response: We have added Supplementary Table 7 that describes the set of *Mla* alleles that do and do not confer resistance to wheat stripe rust.

It appears that the wheat stripe rust resistance of some barley genotypes is based on a single R-gene. Please address this briefly, and comment on the fact that this resistance remains functional. One could have expected it to be broken long time ago.

Response: We have added a paragraph to the Discussion that explores the potential factors that may contribute to maintenance of wheat stripe rust resistance in barley.

Reviewers' Comments:

Reviewer #1:

Remarks to the Author:

I would like to thank the authors for their tremendous efforts in improving their original manuscript, especially for the additional work on generating transgenic barley plants expressing different MLA resistance specificities and for testing for race-specific resistance to the barley powdery mildew fungus in the SxGP DH-47 background. In addition, the text and illustrations have been greatly improved in their clarity for the reader.

However, there are still two important aspects that need to be clarified. Note that I consider this to be an important study that makes claims of far-reaching significance and therefore needs to be supported by irrefutable data sets.

Major points:

1. the new and/or re-organized data in Fig. 4, Fig. 5 and Supplementary Fig. 5 support a major claim by the authors that barley MLA8 is a resistance determinant against wheat rust Pst isolate 16/035. However, the data in Fig. 5A and 5B do not provide conclusive evidence as to whether MLA7 is also a Pst resistance determinant. This claim is based on a single transgenic MLA7 line with multiple T-DNA insertions (T1-4), whereas lines T1-8 and T1-12, which also contain multiple T-DNA insertions, have variable Pst infection phenotypes between 0 and 4. The authors cannot draw firm conclusions about MLA7 as a Pst resistance determinant because the barley line used for transformation, SxGP DH-47, has variable Pst infection phenotypes between 1 and 3 (Fig. 5B lower panel). I suggest that the authors remove the data on the transgenic MLA7 barley plants and the claim that MLA7 detects the presence of Pst. In my opinion, conclusive evidence would require first generating a CRISPR loss-of-function mutant in Golden Promise and then introducing MLA7 as transgene with native 5' and 3' regulatory sequences, which is beyond the scope of this paper.

2. Since the transgenic barley lines expressing different MLA resistance specificities in the S-GP DH-47 background are crucial for the conclusions of this study (Figs. 4 and 5), I suggest that the authors test whether the differential Pst infection phenotypes observed with isolate 16/035 are comparable to that observed with Pst 08/21. This is important because for the latter isolate an extensive data set is shown in Figure 1 and Supplementary Figure 1. This will also directly address my question, raised during evaluation of the first version of this manuscript, as to whether MLA8 detects more than one Pst isolate. Note that Pst 08/21 was used for the QTL mapping shown in Fig. 1. So this will substantiate if Rps7=MLA8. The amount of work seems minimal when limited to the transgenic barley lines expressing MLA8 or lacking the T-DNA insertion.

Minor points

1. to improve the accessibility of the data presented, I strongly suggest that the authors add an additional line in Fig. 5 B ("controls") to indicate the presence or absence of MLA8 in each of the control genotypes, including SusPrit.

2. there are a number of linguistic inaccuracies in the text which I have listed below and which need to be rewritten to improve readability.

Ln 58 to 60. "One of the earliest approaches involved transferring the non-adapted resistance of rye (*Secale cereale* L.) to several pathogens of wheat (*Triticum aestivum* L.) through cytogenetics (chromosome additions)." Please simplify this sentence.

Ln 96 to 99: "Based on this observation, it is unclear whether this indicates that (1) NLRs are extremely specific in their interaction with host proteins such that they have the capacity to only recognize a specific modification or (2) an insufficient number of plant-pathogen pathogen systems have been investigated to establish the broader capacity for NLR recognition." Sentence needs rephrasing.

Ln 415 to 417: "This raises the question, how has barley in the absence of the pathogen, maintained resistance in Australia to Pst despite significant infection pressure in the field?" If you read this sentence carefully, it makes no sense. It needs to be reworded.

Ln 438 to 440: "Taken further, this would suggest that pathogens have a limited number of host targets for suppressing immunity, which are guarded by NLRs, or alternatively, must adopt a conserved structure for effector proteins that is directly recognized by NLRs." This is difficult for the reader to digest. Split this into two sentences and start the second sentence with "In the case of direct detection of effectors by NLRs, ...".

Reviewer #3:

Remarks to the Author:

The manuscript has improved significantly. In fact, it is completely rewritten into another more elaborate format and more data and an explanatory Manhattan plot has been provided.

My question about the contribution of different Mla alleles has been addressed. Pst susceptibility of more transgenic OE lines with more alleles have been generated, and a set of Near-isogenic Mla lines in susceptible Manchuria has been included in the study. This has led to the conclusion that Mla7 and Mla8 provide Pst resistance.

The points that I raised are now discussed satisfactorily.

General Response: Thank you for the careful consideration of the manuscript and edits. Below we detail the modifications we have made principally based on the comments from Reviewer #1.

Reviewer #1 (Remarks to the Author):

I would like to thank the authors for their tremendous efforts in improving their original manuscript, especially for the additional work on generating transgenic barley plants expressing different MLA resistance specificities and for testing for race-specific resistance to the barley powdery mildew fungus in the SxGP DH-47 background. In addition, the text and illustrations have been greatly improved in their clarity for the reader.

However, there are still two important aspects that need to be clarified. Note that I consider this to be an important study that makes claims of far-reaching significance and therefore needs to be supported by irrefutable data sets.

Major points:

1. the new and/or re-organized data in Fig. 4, Fig. 5 and Supplementary Fig. 5 support a major claim by the authors that barley MLA8 is a resistance determinant against wheat rust Pst isolate 16/035. However, the data in Fig. 5A and 5B do not provide conclusive evidence as to whether MLA7 is also a Pst resistance determinant. This claim is based on a single transgenic MLA7 line with multiple T-DNA insertions (T1-4), whereas lines T1-8 and T1-12, which also contain multiple T-DNA insertions, have variable Pst infection phenotypes between 0 and 4. The authors cannot draw firm conclusions about MLA7 as a Pst resistance determinant because the barley line used for transformation, SxGP DH-47, has variable Pst infection phenotypes between 1 and 3 (Fig. 5B lower panel). I suggest that the authors remove the data on the transgenic MLA7 barley plants and the claim that MLA7 detects the presence of Pst. In my opinion, conclusive evidence would require first generating a CRISPR

loss-of-function mutant in Golden Promise and then introducing MLA7 as transgene with native 5' and 3' regulatory sequences, which is beyond the scope of this paper.

Response: We agree with the reviewer that the current evidence does not provide a complete confirmation of *Mla7* as a functional resistance gene to wheat stripe rust. As further work will be substantial and based on the reviewers' view, we have removed this from this current manuscript. To this end, we have modified the text, Fig. 4 and 5 and Supplementary Fig. 4, removed Supplementary Fig. 5, and placed the previous Supplementary Fig. 6 as Fig. 6.

2. Since the transgenic barley lines expressing different MLA resistance specificities in the S-GP DH-47 background are crucial for the conclusions of this study (Figs. 4 and 5), I suggest that the authors test whether the differential Pst infection phenotypes observed with isolate 16/035 are comparable to that observed with Pst 08/21. This is important because for the latter isolate an extensive data set is shown in Figure 1 and Supplementary Figure 1. This will also directly address my question, raised during evaluation of the first version of this manuscript, as to whether MLA8 detects more than one Pst isolate. Note that Pst 08/21 was used for the QTL mapping shown in Fig. 1. So this will substantiate if Rps7=MLA8. The amount of work seems minimal when limited to the transgenic barley lines expressing MLA8 or lacking the T-DNA insertion.

Response: In our efforts on improving the manuscript, we left out the standard differentials used for assessing variation of an isolate. The Manchuria NILs and a collection of SxGP DH lines were tested with *Pst* isolate 16/035 prior to experiments with transgenic lines. No substantial variation was observed relative to *Pst* isolate 08/21, except perhaps for a somewhat more virulent pathotype observed as higher chlorosis levels. These results have now been included as Supplementary Fig. 5 and 6. We would highlight that the previous Supplementary Figure 6 (now Fig. 6), shows that transgenic lines expressing *Mla8* confers resistance to *Pst* isolate 15/151. From the work in Diane Saunder's group at the John Innes Centre, we know that *Pst* isolate 15/151 and 16/035 belong to a genetically similar lineage of wheat stripe rust, which usurped older races in Europe (such as *Pst* isolate 08/21).

Minor points

1. to improve the accessibility of the data presented, I strongly suggest that the authors add an additional line in Fig. 5 B ("controls") to indicate the presence or absence of MLA8 in each of the control genotypes, including SusPrit.

Response: We have edited the figure to include the *Mla* allele and presence of *Rps6*, *Rps7*, and/or *Rps8*. SusPrit and SxGP DH-47 are currently designed as *RGH1.SusPrit*, as it expresses an allele of *Mla* that differs from *Mla13* by two amino acid differences. We have yet to observe avirulence by this genotype to barley powdery mildew or wheat stripe rust.

2. there are a number of linguistic inaccuracies in the text which I have listed below and which need to be rewritten to improve readability.

Ln 58 to 60. "One of the earliest approaches involved transferring the non-adapted resistance of rye (*Secale cereale* L.) to several pathogens of wheat (*Triticum aestivum* L.) through cytogenetics (chromosome additions)." Please simplify this sentence.

Response: This sentence has been modified.

Ln 96 to 99: "Based on this observation, it is unclear whether this indicates that (1) NLRs are extremely specific in their interaction with host proteins such that they have the capacity to only recognize a specific modification or (2) an insufficient number of plant-pathogen pathogen systems have been investigated to establish the broader capacity for NLR recognition." Sentence needs rephrasing.

Response: This sentence has been modified.

Ln 415 to 417: "This raises the question, how has barley in the absence of the pathogen, maintained resistance in Australia to *Pst* despite significant infection pressure in the field?" If you read this sentence carefully, it makes no sense. It needs to be reworded.

Response: This sentence has been modified.

Ln 438 to 440: "Taken further, this would suggest that pathogens have a limited number of host targets for suppressing immunity, which are guarded by NLRs, or alternatively, must adopt a

conserved structure for effector proteins that is directly recognized by NLRs.” This is difficult for the reader to digest. Split this into two sentences and start the second sentence with "In the case of direct detection of effectors by NLRs, ...".

Response: This sentence has been modified.

Reviewer #3 (Remarks to the Author):

The manuscript has improved significantly. In fact, it is completely rewritten into another more elaborate format and more data and an explanatory Manhattan plot has been provided.

My question about the contribution of different Mla alleles has been addressed. Pst susceptibility of more transgenic OE lines with more alleles have been generated, and a set of Near-isogenic Mla lines in susceptible Manchuria has been included in the study. This has led to the conclusion that Mla7 and Mla8 provide Pst resistance.

The points that I raised are now discussed satisfactorily.

Response: No response required.

Reviewers' Comments:

Reviewer #1:

Remarks to the Author:

The authors have satisfactorily addressed my remaining concerns. I appreciate that the claim about Pst resistance mediated by MLA7 has been removed from this re-revised manuscript. The authors present now also convincing evidence that MLA8 can detect at least two Pst isolates, 15/151 and 16/035.

As mentioned in my evaluation of an earlier version of this study, I believe that this work represents an important advance in NLR biology and hope that the journal will find a place to highlight this study via a commentary.

Response to reviewers comments

Reviewer #1 (Remarks to the Author):

The authors have satisfactorily addressed my remaining concerns. I appreciate that the claim about Pst resistance mediated by MLA7 has been removed from this re-revised manuscript. The authors present now also convincing evidence that MLA8 can detect at least two Pst isolates, 15/151 and 16/035.

As mentioned in my evaluation of an earlier version of this study, I believe that this work represents an important advance in NLR biology and hope that the journal will find a place to highlight this study via a commentary.

Response: We thank the reviewer for their effort and positive response to our work. The reviewer did not identify any additional issues, therefore no corrections were needed.